# MiLDEdit: Reasoning-Based Multi-Layer Design Document Editing

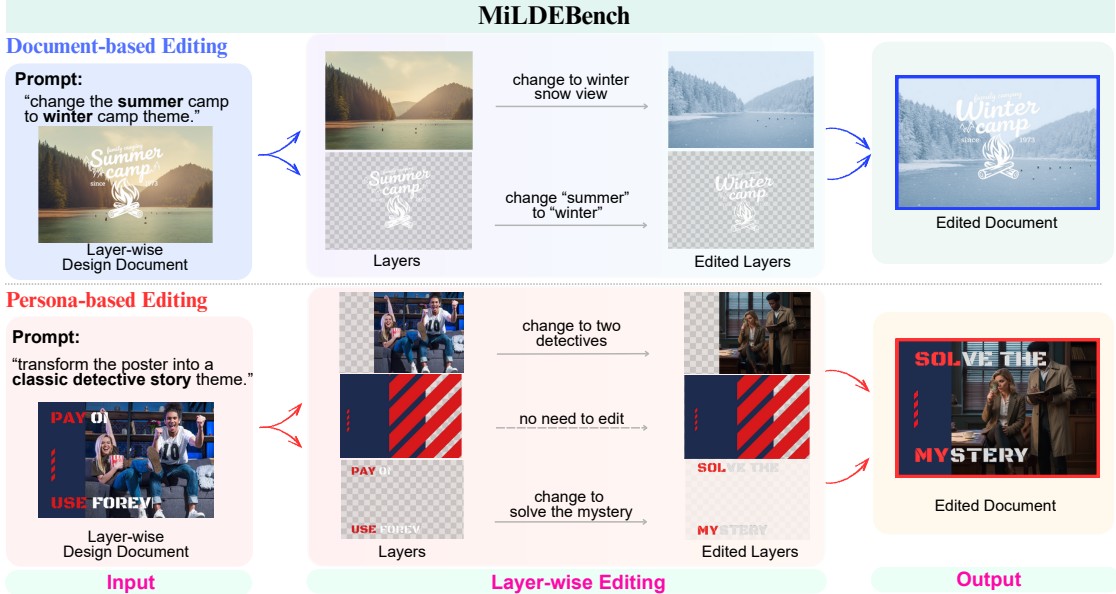

Figure 1: Our benchmark MiLDEBench is the first targeting to transparent-background, multi-layer design document editing.

## Abstract

Real-world *design documents* (e.g., posters) are inherently multi-layered, combining decoration, text, and images. Editing them from natural-language instructions requires fine-grained, layer-aware reasoning to identify relevant layers and coordinate modifications. Prior work largely overlooks *multi-layer design document editing*, focusing instead on single-layer image editing or multi-layer generation, which assume a flat canvas and lack the reasoning needed to determine *what* and *where* to modify. To address this gap, we introduce the Multi-Layer Document Editing Framework (**MiLDEdit**), a reasoning-based framework that combines an RL-trained multimodal reasoner for layer-wise understanding with an image editor for targeted modifications. To systematically benchmark this setting, we introduce the Multi-Layer Document Editing Benchmark (**MiLDEBench**), a human-in-the-loop corpus of over 20K design documents paired with diverse editing instructions. The benchmark is complemented by a task-specific evaluation protocol, **MiLDEEval**, which spans four dimensions including instruction following, layout consistency, aesthetics, and text rendering. Extensive experiments on 13 open-source and 2 closed-source models reveal that existing approaches fail to generalize: open-source models often cannot complete multi-layer document editing tasks, while closed-source models suffer from format violations. In contrast, MiLDEdit achieves strong layer-aware reasoning and precise editing, significantly outperforming all open-source baselines and attaining performance comparable to closed-source models, thereby establishing the first strong baseline for multi-layer document editing.

# 1  Introduction

While recent breakthroughs in image generation have transformed creative workflows, editing real-world design documents such as posters, flyers, and slides still remains an open challenge. Unlike natural images, these design documents are intrinsically multi-layered, combining backgrounds, graphics, text, and foreground imagery in a carefully structured hierarchy. This layered structure is not only a visual property of design documents, but also the native editing representation adopted by widely used design tools, where users modify individual elements rather than a flattened canvas. Therefore, effective document editing naturally requires layer-aware reasoning: the model must identify which layers are relevant to user intent, how their relationships constrain possible modifications, and where changes can be applied without disrupting layout or occluding critical content. Existing reasoning-based editing methods (Jiang et al., 2025; Chen et al., 2025b; Zhang et al., 2025b) are built for flat, single-layer canvases and fail to capture this complexity. Besides, despite some works focusing on design document generation (Huang et al., 2024; Pu et al., 2025; Chen et al., 2025a), layer-aware document editing remains unexplored, leaving a critical gap in vision-language reasoning and multimodal editing.

To fill this gap, we propose the first benchmark for reasoning-based *multi-layer* document editing, Multi-Layer Document Editing Benchmark (**MiLDEBench**). MiLDEBench systematically focuses on *content editing*, which entails semantically coherent modifications while maintaining the visual and structural integrity of the document. Building upon 20K transparent-background templates from the public Crello dataset (Yamaguchi, 2021), we synthesize 50K natural-language editing instructions and 87K layer-aligned edit steps via a hybrid generation pipeline that integrates open-source multimodal LLMs with human-in-the-loop verification. To approximate real-world application scenarios where users come from diverse backgrounds, we design persona-conditioned and document-conditioned prompts that capture heterogeneous editing intents, ensuring that the dataset reflects a broad spectrum of user needs (*e.g.*, converting a Christmas card into a Halloween card). It is worth noting that document editing involves not only content-level modifications but also geometry-level manipulations, such as exchanging the positions of text and image layers in a poster. In this work, we focus on content-aware multi-layer editing, which constitutes a fundamental and challenging step toward controllable document editing. We view geometry-aware editing as an equally important and complementary direction, and leave its systematic exploration to future work.

To evaluate this task, we introduce **MiLDEEval**, a task-specific protocol encompassing four core dimensions: *instruction following*, *layout consistency*, *aesthetics*, and *text rendering*. These dimensions establish a standardized testbed for reasoning-intensive, layer-aware editing that mirrors real-world design scenarios. To better align with human perception, we aggregate these criteria into a unified metric, **MiLDEScore**. This composite score provides a holistic quality assessment and demonstrates a stronger correlation with human preference than previous protocols or individual metrics. We benchmark 13 open-source and 2 closed-source models on MiLDEBench. Given that most existing models produce only flat outputs, we adopt a single-round editing setting where models generate one edited poster from a document and instruction without explicit layer access. Even in this simplified setting, open-source models show limited instruction following, while closed-source models often sacrifice layout consistency for semantic alignment. Notably, incorporating explicit reasoning yields only modest gains, suggesting that current text-centric reasoning fails to leverage multi-layer structures. These findings confirm that design document editing exceeds current paradigms, motivating a specialized reasoning-based, layer-aware approach.

To address these gaps, we propose **MiLDEdit**, a reasoning-based, layer-aware editing framework. MiLDEdit integrates (i) an RL-trained multimodal reasoner, optimized via a novel reward function for layer identification and prompt synthesis, and (ii) a modular editor for targeted, layer-specific modifications. Experimental results show that explicit layer-awareness is crucial for controllable editing. MiLDEdit surpasses open-source baselines by 82.78% in MiLDEScore, achieves performance comparable to closed-source models, and superior layout consistency. By reaching the best balance between *instruction adherence* and *structural fidelity*, MiLDEdit underscores the efficacy of reasoning-based multi-layer editing. Further extensions experiments such as applying a layer decomposition model or specifically finetune the reasoner to adapt one specific image editing model proves the flexibility of MiLDEdit.

We summarize our main contribution as follows:

- **Task and Benchmark.** We formalize the problem of *multi-layer design document editing* and introduce **MiLDEBench**, a corpus of 20K documents with 50K editing instructions and 87K layer-aligned steps, along with the task-specific evaluation protocol **MiLDEEval** and **MiLDEScore**.

- **Comprehensive Evaluation.** We benchmark 13 open-source and 2 closed-source systems, identifying consistent challenges in instruction following, layout fidelity, and coordination across layers.

- **Method and Results.** We propose **MiLDEdit**, which combines a GRPO-trained multimodal reasoner with a pluggable layer-wise editor. MiLDEdit demonstrates strong instruction adherence and layout consistency, surpassing open-source baselines and performing competitively with closed-source systems.

## 2 Related Work

### 2.1 Multi-layer Transparent Image Generation

Prior research on multi-layer design documents has primarily focused on generation and decomposition. Early efforts centered on dataset construction, either by extracting layers from natural image corpora (e.g., LAION (Schuhmann et al., 2022), COCO (Lin et al., 2014)) (Zhang et al., 2023b; Huang et al., 2024; Tudosiu et al., 2024; Gu et al., 2024) or curating professional graphic designs (Yamaguchi, 2021; Pu et al., 2025). Based on these, recent models have advanced multi-layer reasoning (Deng et al., 2025; Xiao et al., 2025b), synthetic supervision pipelines (Burgert et al., 2024; Chen et al., 2025a), and coordinated generation with semantic dependencies (Huang et al., 2024; Pu et al., 2025). Another dimension is layout-related research work that focuses on editing layout (Suri et al., 2024; Avudaiappan & Murali, 2025), decomposing (Nie et al., 2025; Suzuki et al., 2025) or generating layout document or images Feng et al. (2023); Bar-Tal et al. (2022). Our work is different from theirs in that it focuses on reasoning-based, multi-layer transparent image editing.

More recently, research has shifted toward editability via image-to-layer decomposition (e.g., Qwen-Image-Layered (Yin et al., 2025)), enabling post-hoc manual edits. However, these methods primarily focus on layer discovery or reconstruction from flat images, rather than instruction-driven modification of existing design documents. In contrast, real-world workflows require iterative, high-level editing while maintaining structural consistency—a need largely unmet by prior generation-centric work. To bridge this gap, we introduce **MiLDEBench**, the first benchmark pairing layered documents with document-level instructions and human-validated edit traces, shifting the focus to faithful and controllable multi-layer editing.

### 2.2 Reasoning-based Image Generation & Editing

Driven by recent advances in large language models (LLMs) and training algorithms (Shao et al., 2024; Yu et al., 2025b), reasoning-oriented image generation and editing have achieved remarkable progress (Zhang et al., 2025b; Duan et al., 2025; Jiang et al., 2025; Wu et al., 2025b; Guo et al., 2025; Pan et al., 2025; Jin et al., 2024; Zhang et al., 2025a). Current methods may be classified according to the manner in which reasoning is incorporated into the pipeline: (i) *prompt interpretation*, where the system resolves compositional or implicit semantics in user instructions (*e.g.*, temporal or causal cues) prior to editing (Chen et al., 2025b; Sun et al., 2025; Jin et al., 2024; Zhang et al., 2025a); (ii) *prompt extension*, which augments concise instructions with additional structure (*e.g.*, constraints, spatial hints) to enhance output faithfulness (Wu et al., 2025b; Jiang et al., 2025; Zhang et al., 2025b; Duan et al., 2025); and (iii) *generation-time reasoning*, which introduces self-checking or iterative refinement during synthesis to enforce consistency with requirements (Guo et al., 2025; Pan et al., 2025). Nevertheless, these approaches are predominantly built on the assumption of a single, flattened canvas and thus lack *layer-aware* reasoning about hierarchical structure, inter-layer dependencies, and document-level constraints (*e.g.*, text fidelity, non-occluding layout). As a result, even when instructions are correctly interpreted, edits often fail to account for relevant layers or disrupt spatial organization. We introduce **MiLDEdit**, which formalizes *multi-layer document editing* as a reasoning task and ensures consistency via layer selection, layer-wise editing instruction generation, and layer editing.

# 3   MiLDEBench

## 3.1   Preliminaries

We define multi-layer document editing as a two-stage process consisting of reasoning and editing. A document $D$ is represented as an ordered set of transparent layers $\mathcal{L} = \{L_i \in \mathbb{R}^{H \times W \times C}\}_{i=1}^n$, rendered by alpha compositing $D = L_1 \oplus \cdots \oplus L_n$. Given a document-level instruction $I_D$, the reasoning stage is performed by a VLM-based reasoner $\mathcal{R}_\phi(D, I_D) \mapsto \hat{\mathcal{I}} = \{\hat{I}_i\}_{i=1}^n$, which predicts layer-specific instructions where each $\hat{I}_i$ either specifies an edit for layer $L_i$ or is a `no-op` indicating that the layer should remain unchanged. The editing stage is handled by an image-generation editor $\mathcal{E}(\mathcal{L}, I_D, \hat{\mathcal{I}}) \mapsto D'$, which updates the document by applying $L_i' = \mathcal{E}(L_i, \hat{I}_i)$ if $\hat{I}_i \neq$ `no-op`, and $L_i' = L_i$ otherwise. The final edited document is then reconstructed in the original order as $D' = L_1' \oplus \cdots \oplus L_n'$. A valid solution must satisfy *instruction compliance* (the output follows the semantics, text, and attributes of $I_D$), *structural fidelity* (the global layout and all non-target content remain intact), and *layer awareness* (all and only the layers in $S^\star$ are modified). For diagnostic evaluation, the benchmark provides gold supervision in the form of $S^\star$ and $\mathcal{I}$, enabling measurement of both document-level success (instruction following and fidelity) and decision quality (correctness of layer selection and alignment). Each benchmark instance is therefore specified by five components: the rendered document $D$, its layer decomposition $\mathcal{L}$, the document-level instruction $I_D$, the gold relevant-layer set $S^\star$, and the layer-wise instructions $\mathcal{I}$.

Since current open- and closed-source[1] models do not support multi-image (multi-layer) editing interfaces, we design a practical evaluation protocol that treats each method as a *black-box* editor. Specifically, the model only consumes the rendered document $D$ and instruction $I_D$, and produces an edited output $D'$; layer-wise inputs or edits are *not* required. Even under this simplified setting, existing models fail to reliably follow instructions, preserve layout, or render texts (Table 2), underscoring the importance and difficulty of the proposed task: no previous work can fully complete it. Finally, Table 1 summarizes the dataset statistics. We also show the distribution of layers per document and prompt lengths in Figure 6 and 7 in the Appendix.

## 3.2   Dataset Construction Pipeline

The dataset construction pipeline consists of three steps: data collection, document-level instruction generation, and layer-wise instruction generation, with human-in-the-loop validation for the last two steps. Alg. 1 in Appendix A.1 illustrates the overall data creation pipeline. For each generated editing instruction, we ask human annotators to check whether the editing instruction is reasonable based on the design document content. If not, we filter out this example.

Table 1: Statistics of MiLDEBench

| Aspect | Train | Test |
|---|---|---|
| Number of design documents | 17.7k | 1.9k |
| Avg. #layers per doc | 4.45 | 4.44 |
| Avg. #layers needing edit per doc | 1.66 | 1.66 |
| Avg. len of doc-level instruction | 15.56 | 15.53 |
| Avg. len of layer-wise instruction | 24.50 | 24.48 |

**Design document collection and layer consolidation.** We build our corpus from the public Crello dataset (Yamaguchi, 2021), which provides transparent-background, multi-layer *design* documents represented as $(D, \mathcal{L})$, where $D$ is the rendered document and $\mathcal{L} = \{L_i\}_{i=1}^n$ is its layer decomposition. Crello is chosen because (i) our benchmark targets real-world design workflows with non-expert users, so we exclude datasets with synthetically generated layers (*e.g.*, Magick (Burgert et al., 2024), PrismLayers (Chen et al., 2025a)); and (ii) our focus is on scenarios where text, decorative elements, and imagery interact, so we omit multi-layer resources derived from *natural* images (*e.g.*, MuLAn (Tudosiu et al., 2024), MLCID (Huang et al., 2024)). Although ART (Pu et al., 2025) introduces a large-scale design corpus, it is not publicly available and thus excluded. To make $\mathcal{L}$ tractable, we apply a *structure-preserving consolidation* procedure $\mathcal{C}(\mathcal{L}) \mapsto \mathcal{L}'$: an MLLM (InternVL3-38B (Zhu et al., 2025)) classifies layers into *text*, *decoration*, or *image*, and non-overlapping layers within each category are merged using layout metadata while preserving $z$-order and alpha boundaries. This reduces $|\mathcal{L}|$ (2–50) to a semantically coherent $\mathcal{L}'$ ($|\mathcal{L}'|$ varies 1-12) without discarding content.

---

[1]We verified that GPT-o3 could complete the task in manual trials, but the model was discontinued before our benchmark was finalized, preventing systematic evaluation.

**Document-level instruction generation.** Given a consolidated design document $(D, \mathcal{L})$, we generate a document-level instruction $I_D$ for each item. We adopt a two-stream pipeline that balances diversity and realism. (i) *Persona-based stream:* six personas $p_j \sim$ PersonaHub are sampled, and InternVL3-38B generates candidate instructions $I_D^{(j)}$ by adapting $D$ to each persona's domain while preserving its design intent (*e.g.*, "concert poster" → "historical exhibition poster", $p_j$ is a "historian"). (ii) *Document-based stream:* the model proposes semantically proximal domain transfers grounded in $D$ itself (*e.g.*, "summer camp" → "winter camp"). The combined candidate pool $I_D^{(j)}$ is then ranked by clarity, specificity, and realism, with low-quality cases removed through lightweight automatic filtering and regeneration until criteria are met. Finally, a human-in-the-loop validation stage removes instructions that are infeasible, yielding the final $I_D$.

**Layer-wise instruction generation.** For each benchmark instance $(D, I_D, \mathcal{L})$, we provide a set of *layer-aligned* editing instructions $\mathcal{I} = I_i$ specifying how each relevant layer should be modified to realize the document-level intent. During document-level instruction synthesis, the InternVL3-38B is simultaneously prompted to produce step-wise edits as a program that decomposes $I_D$ into atomic actions (*e.g.*, "replace text 'piano concert' with 'historical exhibition'"). We then align steps to layers using a novel MLLM-based content-aware matcher to produce layer-wise instructions $I_i$. The matching algorithm is detailed in Appendix A.2. Finally, automatically generated instructions are filtered by rule-based validators and refined through human-in-the-loop expert review, ensuring clarity, feasibility, and faithfulness to real design workflows. The resulting edited layers $S^\star$ and aligned instructions $\mathcal{I}$ thus combine automated alignment with human refinement to provide reliable gold supervision.

# 4 Benchmarking with MiLDEBench

## 4.1 MiLDEEval

For a comprehensive assessment of our benchmark, we introduce **MiLDEEval**, which encompasses four key evaluation dimensions: instruction following, layout consistency, aesthetics, and text rendering. To holistically reflect model performance on the task, we further integrate the four perceptual criteria into a unified score, denoted as **MiLDEScore**.

**Instruction Following.** To assess whether the model faithfully executes an editing instruction $I_D$, we design a VQA-style evaluation metric. Given the document $D$, the target layer $S^\star$, and its layer-specific prompt $\mathcal{I}$, InternVL3-38B is prompted to generate a question–answer pair for each edited layer. Each question explicitly grounds the edit in spatial, textual, or entity-level detail (*e.g.*, *"Has the main image be changed to a museum scene?"*), with a binary answer of "yes" or "no." The instruction-following score is defined as the proportion of edits judged correct across all layers.

**Layout Consistency.** To evaluate structural fidelity, we measure layout consistency between original and edited documents using mask-level representations. We extract spatial masks $\mathcal{M} = \{M_i\}$ and $\mathcal{M}' = \{M_j'\}$ using Adopd Doc2Mask model (Gu et al., 2024) from the original document $D$ and edited document $D'$, then we design a new matching algorithm to match the two sets of spatial masks. The detailed calculation function is shown in Appendix B.1.

**Aesthetics.** We assess whether edits preserve or improve overall visual appeal using a frozen aesthetics predictor (*Aesthetic Predictor V2.5* (aes)). We directly utilize the score as final evaluation.

**Text Rendering.** We evaluate the *faithfulness* of edited text with an OCR–VQA pipeline. Specifically, we first apply the Adopd Doc2BBox model (Gu et al., 2024) to detect text regions in the edited image $L_j'$, and then use InternVL3-38B to extract the corresponding text $t'$. Given the instruction $I_D$, we prompt the MLLM to verify whether $t'$ satisfies the required edit, producing a score in $\{0, 0.5, 1\}$. Unlike conventional text-alignment metrics (*e.g.*, SentenceBERT (Reimers & Gurevych, 2019)), our approach does not assume a unique ground truth: multiple valid edits may satisfy $I_D$, and thus a judgment-based evaluation better captures instruction faithfulness.

**MiLDEScore.** Although the four evaluation dimensions comprehensively capture different aspects of the multi-layer design document editing task, they cannot be treated as independent objectives. For example, if

an editing model fails to modify the document and simply outputs the unedited input, the *layout consistency* score would reach 100%, while *instruction following* and *text rendering* would be zero. In this case, the high layout consistency is meaningless, since it does not indicate a successful edit. To jointly model the interdependence among these factors, we introduce **MiLDEScore**, a unified metric that aggregates the four perceptual criteria into a single holistic score. Let the raw scores of *instruction following (IF)*, *layout consistency (LC)*, *text rendering (TR)*, and *aesthetics (A)* be normalized to $[0, 1]$ as:

$$\text{IF}_h = \frac{\text{IF}}{100}, \quad \text{LC}_h = \frac{\text{LC}}{100}, \quad \text{TR}_h = \frac{\text{TR}}{100}, \quad \text{A}_h = \frac{\text{A}}{10}. \tag{1}$$

We employ an instruction-following–based **sigmoid gate** to control the influence of other metrics:

$$g(\text{IF}_h) = \frac{\sigma(k(\text{IF}_h - \tau)) - \sigma(-k\tau)}{\sigma(k(1-\tau)) - \sigma(-k\tau)}, \quad \sigma(x) = \frac{1}{1+e^{-x}}, \tag{2}$$

where $\tau$ defines the gate threshold and $k$ controls the steepness. A higher $\tau$ makes the gate stricter, while a larger $k$ sharpens the transition. The overall **MiLDEScore** is computed as:

$$\text{MiLDEScore} = w_{\text{if}}\text{IF}_h + w_{\text{tr}}\text{TR}_h + g(\text{IF}_h)\left(w_{\text{lc}}\text{LC}_h + w_{\text{a}}\text{A}_h\right) + w_{\text{sy}}\, g(\text{IF}_h)\, \text{IF}_h\, \text{LC}_h. \tag{3}$$

The sigmoid gate $g(\text{IF}_h)$ ensures that *layout consistency* and *aesthetics* only contribute meaningfully when the instruction-following score is sufficiently high. When the model fails to follow the editing instruction ($\text{IF}_h < \tau$), the gate value remains near zero, effectively suppressing irrelevant high LC or A scores. As $\text{IF}_h$ increases, these terms are gradually activated, allowing models that both follow instructions and preserve layout to achieve higher overall scores. The last term $w_{\text{sy}}g(\text{IF}_h)\text{IF}_h\text{LC}_h$ captures the synergy between instruction accuracy and spatial consistency. It provides an additional reward when both metrics are simultaneously high, reflecting the natural coupling between semantic correctness and visual coherence in human judgment. More details are discussed in Appendix D.2.

**Layer Decision Accuracy.** In addition to metrics for edited document quality, we also incorporate another metric called layer decision accuracy. As shown in Figure 1, in many cases in our benchmark, not all layers require modification. Therefore, we additionally report the layer decision accuracy to measure whether the model can correctly identify which layers should be edited.

## 4.2 Evaluation and Analysis

To conduct evaluation on **MiLDEBench**, we conduct comprehensive evaluation on 13 open-source models, with 11 reasoning-free models and 2 reasoning-enhanced models, and 2 closed-source models. Note that in these experiments, we only take design document $D$ and document-level editing instruction $I_D$ as input, because current models cannot conduct multiple-layer editing simultaneously. Specifically, the task here is $\mathcal{E}(D, I_D) \rightarrow D'$. The primary results are presented in Table 2. Please refer to Appendix C for detailed experiment and evaluation setup.

**Finding 1: Current image editing models struggle with design document editing.** Our evaluation reveals that both open-source and closed-source models exhibit certain limitations in instruction following and text rendering. For open-source models (`#1`-`#11`), the average instruction-following accuracy is only about 10%, meaning that in nearly 90% of cases the specified edits are not correctly executed. Even the strongest closed-source baseline, GPT-Image-1 (`#14`), achieves only 25.46% instruction following accuracy, underscoring the substantial gap between current image editing capabilities and the demands of multi-layer document editing in realistic scenarios.

**Finding 2: Closed-source models trade layout consistency for instruction following.** Closed-source models (e.g., GPT-Image-1, Nano Banana) significantly outperform open-source counterparts in instruction following, text rendering, and aesthetics. For instance, GPT-Image-1 (`#14`) nearly doubles the instruction accuracy of the best open-source model, Bagel (`#11`) (25.46% vs. 14.23%). Similarly, Nano Banana (`#15`) leads in text rendering (40.32%) over Step1X-Edit (`#12`). However, these gains often come at the cost of layout consistency, where GPT-Image-1 achieves the lowest scores. Notably, the higher consistency scores in

Table 2: Evaluation results of different models. Layer Dec. Accuracy represents layer decision accuracy, respectively. For all scores, higher values indicate better performance. The highest score for closed-source and open-source text-to-image models are marked in red and blue respectively, and underline represents the second in open-source models. Note that for previous baselines incapable of multi-layer editing, the *layer decision accuracy* metric is not applicable.

| # | Model | Instruction Following | Layout Consistency | Aesthetic | Text Rendering | MiLDEScore | Layer Dec. Accuracy |
|---|-------|----------------------|--------------------|-----------|-----------------|------------|---------------------|
| *Open-source Models* | | | | | | | |
| 1 | Instruct-Pix2Pix (Brooks et al., 2023) | 2.30 | 93.46 | 4.23 | 17.16 | 6.23 | – |
| 2 | MagicBrush (Zhang et al., 2023a) | 7.37 | 72.08 | 3.68 | 16.60 | 8.47 | – |
| 3 | UniWorld-v1 (Lin et al., 2025) | 5.75 | 61.59 | 3.91 | 22.04 | 9.15 | – |
| 4 | VAREdit (Mao et al., 2025) | 6.60 | 68.10 | 3.18 | 9.49 | 5.86 | – |
| 5 | UltraEdit (Zhao et al., 2024) | 12.41 | 85.31 | 3.54 | 11.39 | 10.35 | – |
| 6 | AnyEdit (Yu et al., 2025a) | 6.51 | 56.73 | 3.96 | 21.83 | 9.40 | – |
| 7 | OmniGen (Xiao et al., 2025a) | 3.83 | 85.96 | 3.90 | 19.76 | 7.73 | – |
| 8 | Qwen-Image-Edit (Wu et al., 2025a) | 10.09 | 74.20 | 4.12 | 24.32 | 12.42 | – |
| 9 | Flux1-Kontext (Batifol et al., 2025) | 12.49 | 48.32 | 3.94 | 19.31 | 11.58 | – |
| 10 | Step1X-Edit (Liu et al., 2025) | 6.56 | 84.09 | 3.98 | 18.70 | 8.84 | – |
| 11 | Bagel (Deng et al., 2025) | 14.23 | 48.59 | 3.54 | 13.49 | 10.80 | – |
| *Reasoning-enhanced Models* | | | | | | | |
| 12 | Step1X-Edit w/ Thinking | 10.48 | 82.16 | 4.11 | 28.67 | 14.17 | – |
| 13 | Bagel w/ Thinking | 13.60 | 60.91 | 3.65 | 14.51 | 11.23 | – |
| *Closed-source Models* | | | | | | | |
| 14 | GPT-Image-1 (OpenAI, 2025) | 25.46 | 36.24 | 4.66 | 39.67 | 25.60 | – |
| 15 | Nano Banana (DeepMind, 2025) | 24.04 | 58.42 | 4.52 | 40.32 | 27.10 | – |
| *MiLDEdit (Ours)* | | | | | | | |
| 16 | Qwen2.5VL-3B + Flux | 13.29 | 90.15 | 4.32 | 27.52 | 16.10 | 42.90 |
| 17 | Qwen2.5VL-7B + Flux | 20.71 | 93.24 | 4.19 | 36.75 | 25.90 | 80.46 |

Figure 2: (a) Evaluation metrics with editing type. (b) Instruction following score with the number of edited layers. (c) Instruction following the model size.

open-source models frequently stem from "trivial artifacts"—simply outputting the unedited document. This reveals a critical trade-off: while closed-source models follow intent more reliably, they struggle to maintain the structural fidelity essential for real-world design workflows.

**Finding 3: Explicit reasoning mechanisms yield marginal gains for document editing.** Integrating explicit reasoning ("w/ Thinking") into open-source editors provides limited improvements. While Step1X-Edit (#10) shows a modest boost in instruction accuracy (from 6.56% to 10.48%), Bagel (#11), conversely, shows a slight performance decline. These results suggest that current reasoning modules primarily capture textual intent but fail to ground edits within complex multi-layer structures, especially when simultaneous text and image modifications are required. This underscores the need for deep multimodal reasoning integration rather than shallow textual planning to truly advance document editing capabilities.

**Finding 4: Performance degrades sharply as editing complexity and layer depth increase.** Model efficacy is highly sensitive to task complexity across modalities and structure. As shown in Figure 2 (a), instruction following drops significantly when transitioning from single-modality edits (e.g., 13.7% for

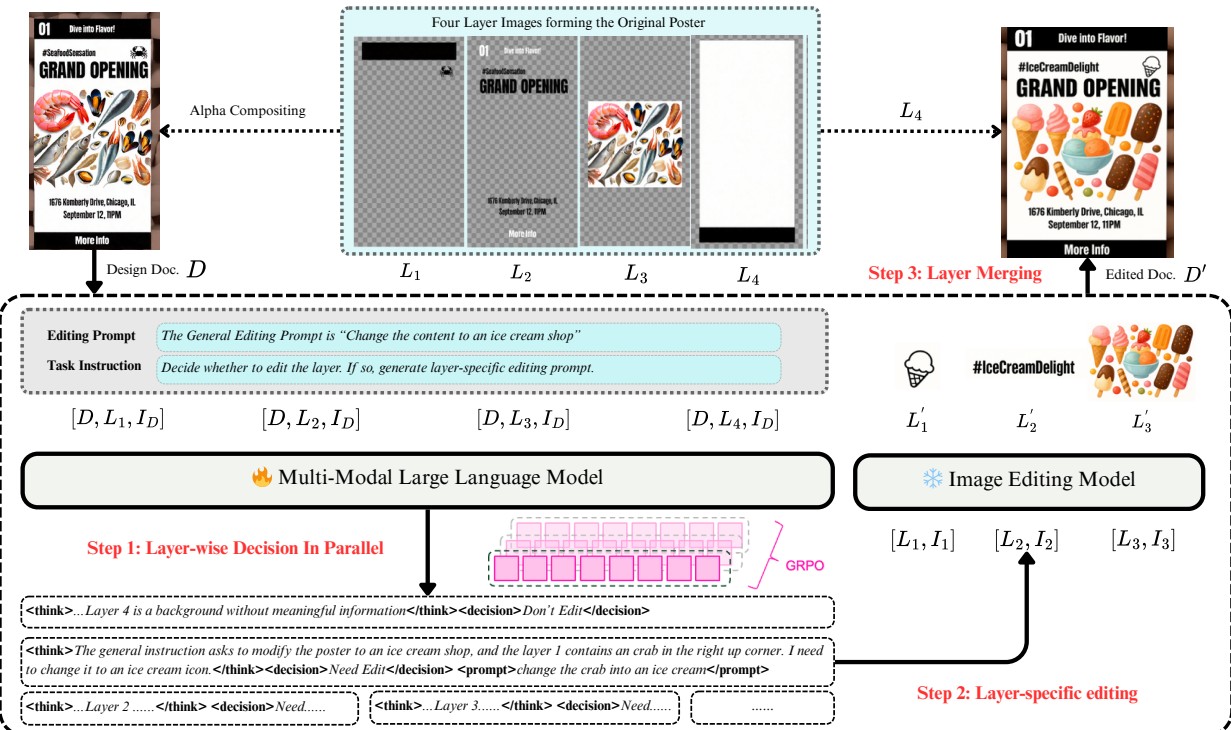

Figure 3: The illustration of MiLDEdit.

text-only) to combined text+image tasks (7.6%). Furthermore, Figure 2 (b) demonstrates that layer depth poses a major challenge: Bagel's accuracy falls from 20.1% (one layer) to 10.6% (three layers), a trend mirrored by even the strongest closed-source models. Finally, 2 (c) shows that larger model size does not consistently yield improvements, suggesting that the bottleneck lies in multimodal reasoning capability rather than pure parameter scaling. These findings highlight the difficulty current models face in navigating complex, multi-layered editing intents.

## 5    The MiLDEdit Framework

Recognizing the reasoning inaccuracies, layout consistency issue and the fundamental problem that current image editing model cannot do multiple layer editing, we propose **MiLDEdit**, consisting of an RL-trained reasoner and a frozen image editor. Specifically, our agent receives a design document $D$ with multiple transparent background layers $\mathcal{L}$ and a document-level instruction $I_D$, and then produce $D'$ by editing *exactly* the relevant layers and re-compositing them in the original $z$-order. Specifically, the task here is $Agent(D, I_D, \mathcal{L}) \rightarrow (D', \mathcal{L}')$. We introduce our agent in Section 5.1 and evaluate on our benchmark, including ablation studies and human evaluation, on Section 5.2.

### 5.1    Reasoning-Guided Multi-Layer Document Editing

As shown in Figure 3, our **MiLDEdit** is a two-stage framework for multi-layer document editing, where the reasoner $\mathcal{R}_\phi$ performs instruction decomposition and the editor $\mathcal{E}$ performs layer-wise editing.

**Reasoning.** The reasoning stage is handled by a VLM-based reasoner $\mathcal{R}_\phi$, which takes $(D, L_i, I_D)$ as input and outputs for each layer a binary decision $y_i \in \{0, 1\}$ and, if $y_i = 1$, a *layer-conditioned prompt* $I_i$. To train $\mathcal{R}_\phi$, we adopt Group Relative Policy Optimization (GRPO) (Shao et al., 2024), a RL method that evaluates groups of sampled responses, computes relative advantages by normalizing their rewards, and applies a clipped KL-regularized objective. This design reduces variance in credit assignment and encourages

the model to distinguish between relatively better and worse responses, which is particularly beneficial for structured reasoning tasks (see Appendix D.1 for details).

Following this paradigm, we design a task-specific per-layer reward to supervise $\mathcal{R}_\phi$. The outputs of the reasoner must follow a structured format:

$$\texttt{<think>...</think><decision>...</decision><prompt>...</prompt>} \tag{4}$$

where the three segments denote hidden reasoning, the binary decision $y_i$, and the layer-conditioned prompt $I_i$, respectively. The per-layer reward $\mathcal{R}_i$ then consists of three components:

$$r_f = \mathbb{1}[\text{format is valid}], \quad r_d = \mathbb{1}[y_i = y_i^\star], \quad r_p = \text{BLEU}(I_i, I_i^\star) \in [0, 1]. \tag{5}$$

The final per-layer reward is defined as

$$\mathcal{R}_i = \begin{cases} (r_f + r_d + r_p)/3, & r_d = 1, \\ (r_f + r_d)/2, & r_d = 0 . \end{cases} \tag{6}$$

where $r_f$ verifies syntactic correctness, $r_d$ measures decision accuracy against the gold label $y_i^\star = \mathbb{1}[L_i \in S^\star]$, and $r_p$ evaluates prompt quality relative to the reference instruction $I_i^\star$. The prompt reward $r_p$ is only applied when the decision is correct ($r_d = 1$).

**Editing.** The editing stage uses a frozen image-generation editor $\mathcal{E}$ for stability and modularity. For each selected layer $L_i$ ($y_i = 1$), a binary mask $M_i$ is extracted from its alpha channel (optionally refined with region cues), and the editor updates it as $L_i' = \mathcal{E}(L_i, I_i, M_i)$. For non-selected layers ($y_i = 0$), no operation is applied and $L_i' = L_i$. Transparency is preserved by restoring the original alpha to unedited regions. Please note that we do not allow changes to geometry (e.g., text reshaping or object movement) as our task asks the model to preserve the original format. We acknowledge that in some cases, it is necessary to change the text shaping, and we leave such situations as future work. The final document is reconstructed by alpha compositing $D' = L_1' \oplus L_2' \oplus \cdots \oplus L_n'$, where $\oplus$ denotes standard alpha blending, ensuring global layout consistency while fulfilling the document-level instruction $I_D$.

## 5.2 Experimental Results

**Setup.** We incorporate one of the SOTA MLLM, `QwenVL2.5-3B/7B` (Bai et al., 2025) as our reasoner, and applied the GRPO algorithm to train on content editing tasks, with a frozen `Flux-1-Kontext` as editing model. The rollout number is 5 and the batch size is 512. Experiments are conducted on 8 A100 GPUs.

**Quantitative Results.** As shown in Table 2, our proposed MiLDEdit substantially outperforms all baselines in the content editing regime. MiLDEdit achieves 25.9% in MiLDEScore an 82.78% improvement over the strongest open-source baseline (Bagel, 14.17%), narrowing the gap with closed-source systems (Nano Banana, 27.1%) and even surpassing GPT-Image-1 (25.6%) while preserving editability. Beyond overall editing quality, the 7B MiLDEdit also outperforms all open-source baselines on instruction following, and maintains strong layout consistency at 93.2%, matching the best diffusion-based editors and exceeding closed-source models by over +30 points. Notably, our agent shows markedly stronger text rendering (36.8%), outperforming all open-source baselines ($\leq 24.3\%$) and approaching commercial systems (40%), which underscores the advantage of our reasoning-based design for multi-layer textual elements. Finally, MiLDEdit achieves 80.5% layer decision accuracy, demonstrating robust layer-aware reasoning that is entirely absent from existing baselines and validating the necessity of reasoning-enhanced frameworks for this task. Taken together, these results indicate that multi-layer document editing benefits from explicit reasoning mechanisms rather than relying solely on generation or editing heuristics: MiLDEdit consistently balances instruction fidelity, fine-grained text rendering, and layer-aware decomposition, making it the first system to robustly address multi-layer editing at scale.

**Quality Analysis.** As illustrated in Figure 4, the input design document consists of three layers. Our agent successfully identifies that the first layer, which contains the background image of Los Angeles, should be edited to depict New York City. In addition, the text "in Los Angeles" in the second layer is correctly

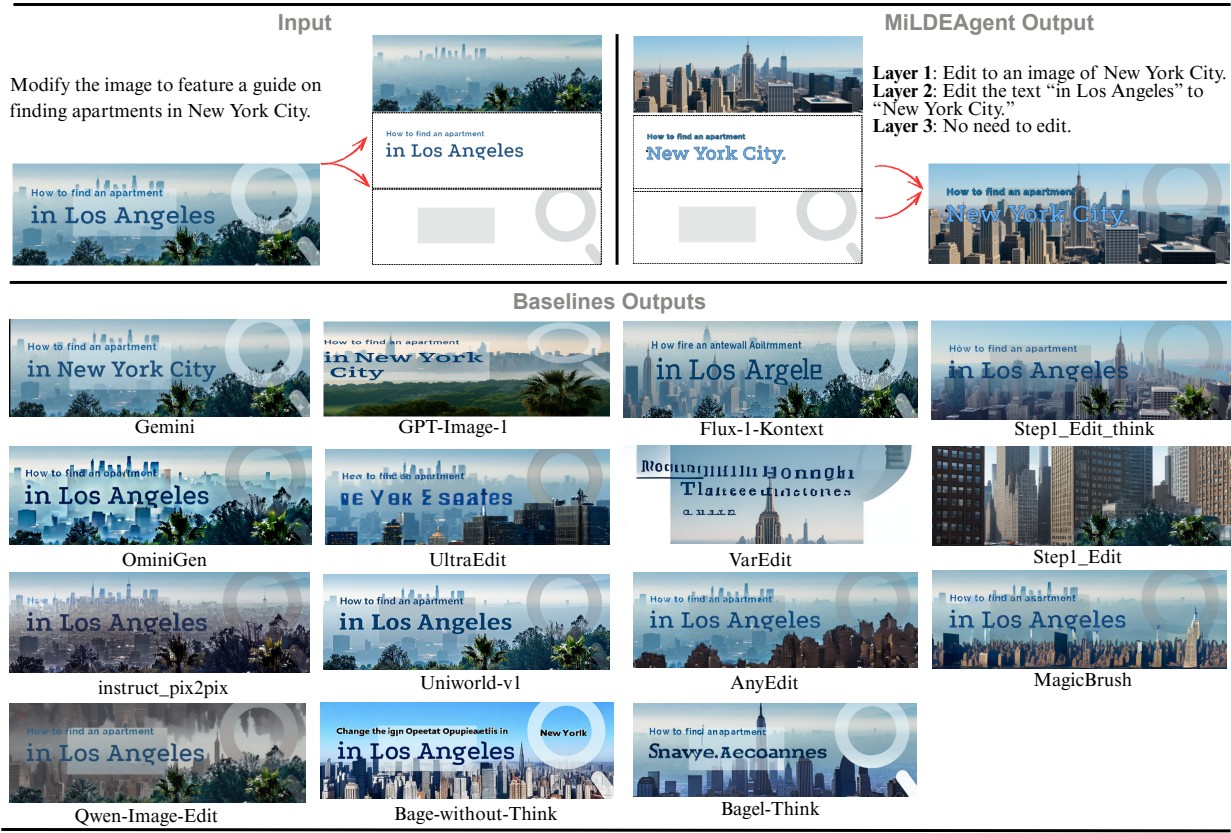

Figure 4: Qualitative comparison results between MiLDEdit and other baselines. The input consists of one user's query and three layers, which compose the original design document. MiLDEdit's output includes layer decisions and edited layers, which compose the final edited design document.

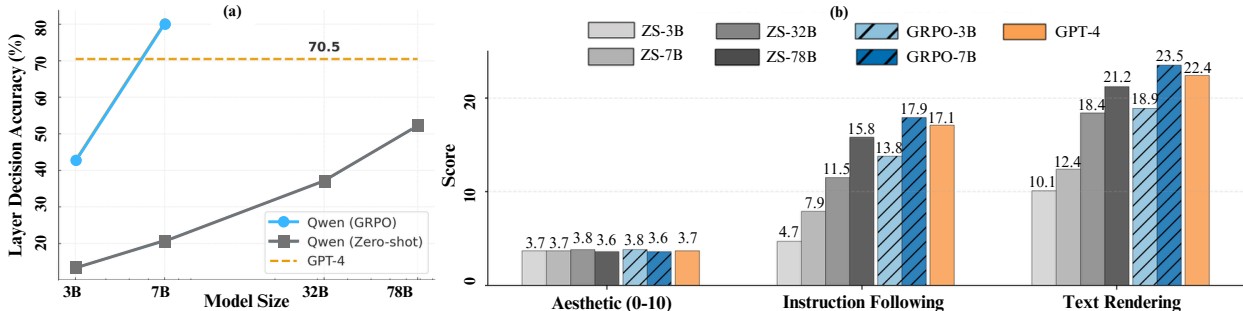

Figure 5: (a) Results of layer decision accuracy versus model sizes. (b) Evaluation metrics with different reasoning models. ZS-xB represents the reasoning models without training, while GRPO-xB represents the GRPO-trained reasoning models. We choose Flux1-Kontext as the editor.

modified to "New York City." The third layer, however, is purely decorative and is correctly recognized as not requiring any modification. After applying an open-source image editing model, our agent composites the edited layers with the unedited ones to form the final output. The resulting image preserves the original layout while accurately updating the relevant content, and it also retains per-layer information for future flexible modifications by users. In contrast, all other baselines fail in this task, even under single-image editing settings. For instance, Gemini only changes the textual content without modifying the background image, whereas GPT-Image-1 fails to maintain layout consistency. Other open-source baselines either fail to edit the text (e.g., OmniGen, Step1-Edit) or completely fail to perform meaningful edits (e.g., VarEdit).

Table 4: Human evaluation on **MiLDEBench** (left) and **Inter-Annotator Agreement** (right, Cohen's $\kappa$). IF, LC, Aes. TR, OQ represent instruction following, layout consistency, aesthetic, text rendering, and overall quality, respectively.

| # | Model | IF | LC | Aes. | TR | OQ |
|---|---|---|---|---|---|---|
| *Baselines* | | | | | | |
| 1 | Instruct-Pix2Pix | 0.05 | 2.85 | 1.23 | 0.67 | 0.27 |
| 2 | MagicBrush | 0.35 | 2.27 | 1.07 | 0.59 | 0.28 |
| 13 | Bagel w/ Thinking | 0.87 | 2.05 | 1.16 | 0.58 | 0.39 |
| 15 | Nano Banana | 1.34 | 1.76 | 1.95 | 1.84 | 1.45 |
| *MiLDEdit (Ours)* | | | | | | |
| 17 | Qwen2.5VL-7B + Flux | 1.28 | 2.83 | 1.27 | 1.75 | 1.37 |

| Metric | $\kappa$ |
|---|---|
| IF | 0.75 |
| LC | 0.71 |
| Aes. | 0.61 |
| TR | 0.72 |
| OQ | 0.69 |
| AVG | 0.70 |

**Ablation 1: GRPO-trained reasoner outperforms all zero-shot models in layer decision accuracy.** Reasoner is the key to MiLDEdit, therefore, we conduct an ablation study on the RL-trained reasoner with other larger open-/closed-source MLLMs on layer decision accuracy metrics. As shown in Figure 5 (a), we observe that models equipped with a GRPO-trained reasoner consistently surpass their zero-shot counterparts across all tested scales. For instance, QwenVL2.5-7B with GRPO achieves 80.5% accuracy, compared to only 20.7% for its zero-shot variant, a nearly $4\times$ improvement, and outperforms GPT-4 by 13.6%, highlighting that structured reinforcement-style reasoning is beneficial even at smaller scales. We further randomly select 100 samples to test the final editing. As shown in Figure 5 (b), using the GRPO-trained 7B model outperforms GPT-4 in both instruction following and text rendering, which is consistent with the layer decision accuracy results. These results underscore that reasoning-oriented training is the dominant factor for reliable layer decision making, establishing GRPO as a crucial ingredient for advancing multi-layer document editing.

**Ablation 2: Image editing model also influence the final performance.** In this experiment, we randomly select 100 samples from the content editing test set and utilize the GRPO-trained QwenVL2.5-7B model as reasoner to test different image editing models. As shown in Table 3, GPT-Image-1 consistently achieves the best overall results, with 19.7% in instruction following, 4.4% in aesthetics, and 30.6% in text rendering, outperforming the best open-source alternatives by a clear margin. Among open-source models, Qwen-Image-Edit exhibits relatively stronger instruction following and text rendering, while Bagel and Flux1-Kontext are more balanced but weaker in fidelity and reasoning. These results indicate that even with

Table 3: Evaluation score with different image editing models.

| Model | IF | Aes. | TR |
|---|---|---|---|
| Qwen-Image-Edit | 18.5 | 4.0 | 26.4 |
| Flux-1-Kontext | 17.9 | 3.65 | 23.5 |
| Bagel | 17.3 | 3.76 | 25.4 |
| GPT-Image-1 | 19.7 | 4.37 | 30.6 |

the same reasoning mechanism, the fidelity and controllability of the editing backbone strongly shape the final quality of document editing. Finally, we acknowledge that using a more powerful image editing model will improve the performance, but this will not influence the main conclusion of our findings.

**Human Evaluation.** We randomly sample 100 instances from **MiLDEBench** for human evaluation. To ensure representative comparison while controlling annotation cost, we include four baselines: Nano Banana (closed-source), Bagel w/ Thinking (reasoning-enhanced), and Instruct-Pix2Pix and MagicBrush (open-source). We evaluate the Qwen2.5VL-7B–based MiLDEdit yielding 500 edited outputs (5 methods $\times$ 100 instances), each independently assessed by two annotators. Annotators consist of Ph.D. or master's students with expertise in multimodal learning, as well as professional designers experienced in document layout and visual composition. Following the protocol of MiLDEBench each output is rated on instruction following, layout consistency, aesthetic quality, and text rendering, together with an overall quality score. We adopt a discrete scale of $0, 1, 2, 3$ for efficiency, where 0 indicates failure and 1–3 correspond to poor, fair, and good quality, respectively. The aggregated results are reported in Table 4. Human judgments are consistent

with the automatic metrics in Table 2: MiLDEdit performs comparably to the closed-source Nano Banana and significantly outperforms open-source baselines across most dimensions. The average inter-annotator agreement (IAA) is approximately 0.7, indicating substantial agreement and supporting evaluation reliability; agreement on aesthetic quality is relatively lower due to its inherently subjective nature.

## 6 MiLDEdit Extensions

### 6.1 Extension to Automatically Decomposed Layers

This work focuses on a structured document editing setting where the input document is represented as a set of transparent layers. In these workflows, users typically manipulate individual design elements rather than editing the entire document as a flat raster image. At the same time, we acknowledge that original layer files may not always be available. For example, a user may start from a flat image generated by a text-to-image model, a screenshot, or a rasterized poster downloaded from the web. To evaluate whether our framework can be extended to such cases, we use Qwen-Image-Layered (QIL) (Yin et al., 2025) as an automatic decomposition module. Specifically, we randomly sample 100 examples, decompose each flat input image into layers using QIL, and then apply our reasoning-guided multi-layer editing framework without any retraining or architectural modification. We evaluate three decomposition settings: the default 4-layer output of QIL, an over-decomposition setting with 10 layers, and an oracle setting where QIL is given the ground-truth number of layers. We utilize Flux-1-Kontext as the image editing model.

As shown in Table 5, our framework remains effective when layers are automatically recovered. With original layers, MiLDEdit achieves 17.9 instruction following, 94.7 layout consistency, and 23.5 text rendering. Using QIL with its default 4-layer setting still obtains 16.8, 92.9, and 20.7, respectively, while the oracle-number setting further improves to 17.1, 93.4, and 21.6, approaching the performance with original layers. In contrast, the 10-layer setting drops to 11.2, 87.5, and 16.8 because over-decomposition often introduces redundant or duplicated layers, such as splitting the same text element into several visually similar layers with different opacity or grayscale values. Since MiLDEBench contains 4.4 layers on average, the 4-layer and oracle settings better match the dataset statistics and preserve most of the benefits of layer-aware editing. These results demonstrate that our framework can naturally extend to realistic scenarios where only a flat document image is available, and its performance improves as the recovered layer structure becomes more accurate.

| Model | Instr. Following | Layout Consist. | Text Rendering |
|---|---|---|---|
| MiLDE (7B)-Flux | 17.9 | 94.7 | 23.5 |
| QIL (4 layers) | 16.8 | 92.9 | 20.7 |
| QIL (10 layers) | 11.2 | 87.5 | 16.8 |
| QIL (Oracle) | 17.1 | 93.4 | 21.6 |

Table 5: Results with automatically decomposed layers on 100 sampled examples. "Oracle" denotes using the ground-truth number of layers for QIL decomposition.

### 6.2 Extension to Specific Editing Model

In our primary framework design, the reasoning module is trained in an *editor-agnostic* manner. Specifically, it learns to generate layer-aware editing prompts without explicit conditioning on or knowledge of the downstream image editor's architecture or instruction-following behaviors. This intentional decoupling stems from the practical reality that different image-editing models exhibit highly diverse prompting preferences and architectural biases. Our core objective is to establish a generalized, highly transferable reasoning front-end that seamlessly pairs with arbitrary downstream editors, rather than overfitting to the quirks of a single specific model. However, we acknowledge that such generality can introduce a performance gap when deployed with a specific downstream editor. To systematically investigate this behavior and analyze the impact of editor-specific optimization, we introduce an *editor-in-the-loop* training variant using `Flux-1-Kontext` as the target editing model.

For each instance in the training set, we generate five distinct candidate editing prompts using an LLM. We then execute `Flux-1-Kontext` to modify the designated layers using each candidate prompt independently. To close the feedback loop, we employ `Qwen-3VL-32B` as an automated judge to evaluate the edited outputs

and select the prompt that exhibits the highest instruction-following fidelity. Finally, we retrain our reasoning module from scratch using these selected, editor-optimized prompts as the new ground-truth training targets. We denote this specialized model variant as $\text{MiLDE}_{\text{Flux-Specific}}$.

Table 6: Cross-editor evaluation comparing the original framework ($\text{MiLDE}_{\text{General}}$) with the specialized (editor-in-the-loop) variant.

| Reasoner | Editor | Instruction Following | Layout Consistency | Aesthetic | Text Rendering | MiLDEScore |
|---|---|---|---|---|---|---|
| $\text{MiLDE}_{\text{General}}$ (Ours) | Flux-1-Kontext | 20.71 | **93.24** | 4.19 | 36.75 | 25.94 |
| $\text{MiLDE}_{\text{General}}$ (Ours) | Qwen-Image-Edit | 19.12 | 92.86 | 4.21 | 34.25 | 23.92 |
| $\text{MiLDE}_{\text{Flux-Specific}}$ | Flux-1-Edit | **22.64** | 93.15 | **4.31** | **39.65** | **26.73** |
| $\text{MiLDE}_{\text{Flux-Specific}}$ | Qwen-Image-Edit | 17.30 | 91.31 | 4.26 | 35.88 | 23.50 |

The empirical results of this cross-evaluation are detailed in Table 6. When evaluating on the targeted editor (`Flux-1-Edit`), the editor-specific variant ($\text{MiLDE}_{\text{Flux-Specific}}$) yields noticeable improvements over the generalized baseline, elevating the overall MiLDEScore from 25.90 to 26.70. This boost is predominantly driven by enhanced instruction following ($+1.93$) and superior text rendering capabilities ($+2.90$), demonstrating that targeted data filtering aligns the reasoner tightly with the preferences of the downstream editor. Conversely, a severe drop in performance is observed when the same Flux-optimized reasoner is transferred to an unseen editor, `Qwen-Image-Edit`. Here, the overall MiLDEScore degrades from 26.73 to 23.50, with a particularly stark drop in the Instruction Following metric (from 22.64 down to 17.30).

These dynamics highlight a critical architectural trade-off: while editor-in-the-loop training successfully capitalizes on model-specific biases to maximize local performance, it simultaneously compromises cross-editor transferability and generalizability. Based on these findings, we retain the editor-agnostic pipeline as our default configuration to ensure broad applicability, while offering editor-specific tuning as a complementary, high-performance option when the downstream deployment pipeline is fixed.

# 7 Limitations and Future Work

Although our approach achieves state-of-the-art performance among open-source image editing models and obtains competitive results compared with closed-source systems, it still has several limitations.

First, our current task formulation focuses on content-level layer editing and does not cover geometry-changing operations. For example, edits that require moving, resizing, reshaping, or swapping the positions of text and image layers are outside the scope of the current benchmark and method. We believe that extending multi-layer document editing to support such geometric transformations is an important direction for future work.

Second, our current framework does not fully resolve cross-layer editing conflicts. As illustrated in Appendix G, modifying one layer may affect the visual compatibility or editing decision of other layers after recomposition. Future work could incorporate joint reasoning over multiple layers, post-recomposition verification, and training objectives that explicitly account for foreground-background compatibility, text readability, and global design coherence.

Finally, as we discussed in Sec. 3, our proposed dataset, MiLDEBench, does not contain ground truth edited image. Future work may propose a novel way to get reliable ground truth edited documents for finetuning specific layer-wise image editing models.

# 8 Conclusion

In this work, we introduced MiLDEBench, the first benchmark for reasoning-based multi-layer poster editing, together with a novel evaluation metric. Through comprehensive experiments, we demonstrated that existing methods struggle to accurately edit posters based on general simple editing prompt. To address these limitations, we proposed MiLDEdit, leveraging a GRPO-trained reasoner for layer selection, coupled with an open-source image editor, significantly improving reasoning ability and editing quality.

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

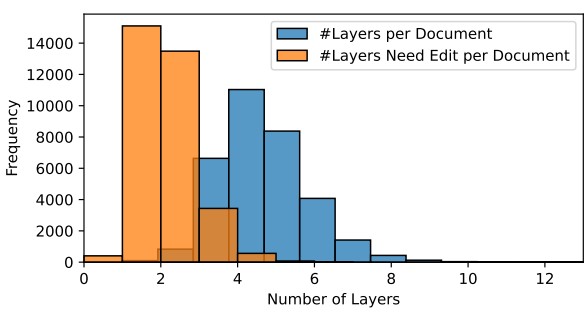

(a) Train set #layer distribution.

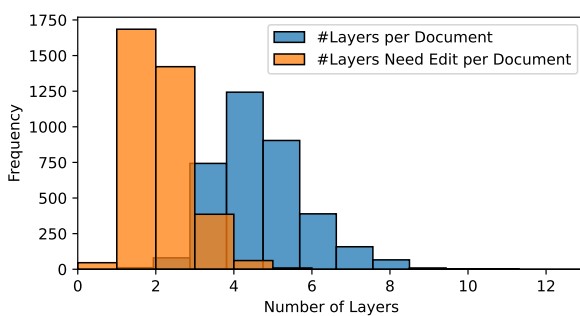

(b) Test set #layer distribution.

Figure 6: Distributions of the total number of layers per document and the number of layers requiring edits per document in the MiLDEBench.

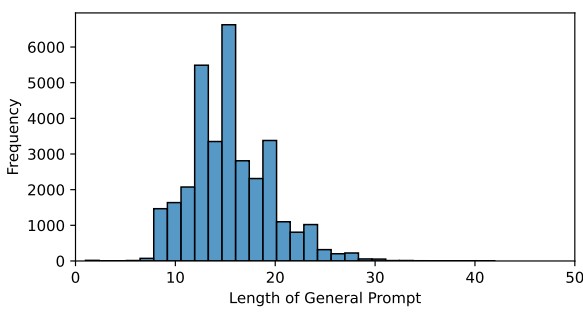

(a) Distribution of general prompt length.

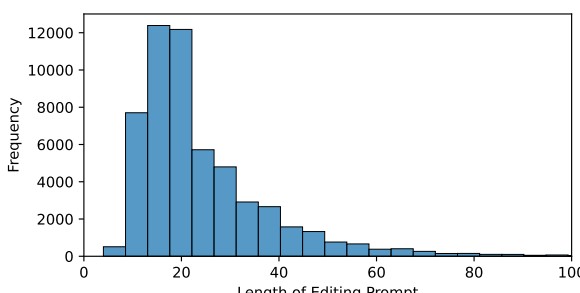

(b) Distribution of editing prompt length.

Figure 7: Distributions of general prompt lengths and the editing prompt lengths in the MiLDEBench.

# A  MiLDEBench

## A.1  Data Generation Pipeline

---

**Algorithm 1:** Data Construction Pipeline

**Input**    : Design document $D$ with layers $\mathcal{L}$
**Output** : Validated document-level instruction $I_D$, layer-wise instructions $\mathcal{I} = \{I_i\}$, edited layers $S^{\star}$

**Part A: Document-level Instruction Generation**
  1. Generate candidate instructions $\{I_D^j\}$ from $D$ via personas $p_j \sim$ PersonaHub;
  2. Rank and filter $\{I_D^j\}$ by clarity, realism, and consistency;
  3. Human validation $\Rightarrow$ finalize $I_D$.

**Part B: Layer-wise Instruction Generation**
  1. Decompose $I_D$ into step-wise edits $\mathcal{A} = \{a_j\}$;
  2. Match each $a_j$ to candidate layers $L_k \in \mathcal{L}$ using content-aware alignment;
  3. Form preliminary instructions $I_k$ and filter by clarity, feasibility, and consistency;
  4. Human validation $\Rightarrow$ finalize $\mathcal{I}$ and relevant-layer set $S^{\star}$.

---

## A.2  Layer-wise Instruction Generation

In this section, we describe the matcher used to align step-wise editing prompts with document layers. Given a set of step-wise prompts $\mathcal{I}_k$ and the layer set $S_j$ with known types (textual or visual), we first classify each prompt $\mathcal{I}_k$ using InternVL3-38B into either a *text-editing* or an *image-editing* category. A prompt is

considered eligible only for layers of the corresponding type (i.e., text prompts for textual layers, image prompts for visual layers). Within each category, we process prompts sequentially: for each $\mathcal{I}_k$, we traverse the candidate layers in $z$-order and query InternVL3-38B to assess whether $\mathcal{I}_k$ semantically applies to $S_j$. Upon a positive match, $\mathcal{I}_k$ is assigned to $S_j$, and the procedure advances to the next prompt. This iterative matching continues until all prompts have been assigned or no valid layer remains.

## B MiLDEEval

### B.1 Layout Consistency

To evaluate structural fidelity, we measure layout consistency between original and edited documents using mask-level representations. We extract spatial masks $\mathcal{M} = \{M_i\}$ and $\mathcal{M}' = \{M_j'\}$ using Adopd Doc2Mask model (Gu et al., 2024) from the original document $D$ and edited document $D'$, then we design a new matching algorithm to match the two sets of spatial masks. For matched pairs, we assess position consistency (normalized centroid displacement), shape consistency (IoU), and area consistency (size ratio). Unmatched layers incur area-proportional penalties, with deleted layers penalized more heavily than newly created ones. The final score combines matching rate, average consistency scores, and penalty deductions with empirically tuned weights, providing a comprehensive measure of layout preservation robust to structural variations.

To assess the **structural fidelity** requirement—specifically whether the edited document $D'$ preserves the spatial arrangement and geometric relationships of elements—we introduce a comprehensive layout consistency metric that operates on mask-level representations of document layers. Given the inherent challenges of multi-layer editing where the number of layers may change ($|\mathcal{L}'| \neq |\mathcal{L}|$) and layer correspondences may be disrupted due to editing operations, our evaluation framework employs a principled matching strategy followed by multi-dimensional consistency assessment.

**Mask Extraction and Matching.** For both the original document $D$ and edited document $D'$, we extract layer-wise masks $\mathcal{M} = \{M_i\}_{i=1}^{|\mathcal{L}|}$ and $\mathcal{M}' = \{M_j'\}_{j=1}^{|\mathcal{L}'|}$ respectively using Adopd Doc2Mask model Gu et al. (2024), where each mask $M_i \in [0,1]^{H \times W}$ represents the spatial footprint of layer $L_i$. To establish correspondences between original and edited layers, we formulate mask matching as a bipartite graph optimization problem: we compute a pairwise IoU similarity matrix $\mathbf{S} \in \mathbb{R}^{|\mathcal{L}| \times |\mathcal{L}'|}$ where $S_{ij} = \text{IoU}(M_i, M_j')$, then apply the Hungarian algorithm to find the optimal matching $\mathcal{P}^* = \arg\max_{\mathcal{P}} \sum_{(i,j) \in \mathcal{P}} S_{ij}$ subject to IoU threshold filtering ($S_{ij} \geq \tau_{\text{IoU}}$).

**Multi-Dimensional Consistency Assessment.** For each matched pair $(M_i, M_j') \in \mathcal{P}^*$, we evaluate three complementary aspects of layout preservation: (1) **Position consistency** measures centroid displacement normalized by image diagonal: $c_{\text{pos}}(M_i, M_j') = 1 - \frac{\|\text{centroid}(M_i) - \text{centroid}(M_j')\|_2}{\sqrt{H^2 + W^2}}$; (2) **Shape consistency** directly uses the IoU between masks: $c_{\text{shape}}(M_i, M_j') = \text{IoU}(M_i, M_j')$; (3) **Area consistency** computes the ratio of smaller to larger mask areas: $c_{\text{area}}(M_i, M_j') = \frac{\min(\text{area}(M_i), \text{area}(M_j'))}{\max(\text{area}(M_i), \text{area}(M_j'))}$.

**Unmatched Layer Penalty.** To account for layers that appear or disappear during editing, we introduce a penalty mechanism that distinguishes between disappeared layers (present in $\mathcal{L}$ but unmatched in $\mathcal{L}'$) and newly created layers (present in $\mathcal{L}'$ but unmatched in $\mathcal{L}$). The penalty for each unmatched layer is proportional to its normalized area, with disappeared layers receiving full penalty and new layers receiving a reduced penalty (coefficient 0.7) to reflect that layer creation may be intentional: $p_{\text{disappeared}} = \sum_{i \in \mathcal{U}_{\text{orig}}} \text{area}(M_i)$ and $p_{\text{new}} = 0.7 \sum_{j \in \mathcal{U}_{\text{edit}}} \text{area}(M_j')$, where $\mathcal{U}_{\text{orig}}$ and $\mathcal{U}_{\text{edit}}$ denote unmatched layer indices.

**Final Score Computation.** The overall layout consistency score aggregates matched-layer performance with unmatched-layer penalties:

$$
\begin{aligned}
\text{LayoutConsistency} = \max\Big(&0, \omega_{\text{match}} \cdot r_{\text{match}} + \omega_{\text{pos}} \cdot \bar{c}_{\text{pos}} + \omega_{\text{shape}} \cdot \bar{c}_{\text{shape}} \\
&+ \omega_{\text{area}} \cdot \bar{c}_{\text{area}} - \omega_{\text{penalty}} \cdot (p_{\text{disappeared}} + p_{\text{new}})\Big),
\end{aligned}
\tag{7}
$$

Table 7: Introduce of each baseline model. Rea.-En. represents whether the model is reasoning-enhanced.

| Model | Size | Type | Rea.-En. |
|---|---|---|---|
| Instruct-Pix2Pix Brooks et al. (2023) | 1B | Diffusion | ✗ |
| MagicBrush Zhang et al. (2023a) | 1B | Diffusion | ✗ |
| UniWorld-v1 Lin et al. (2025) | 20B | Diffusion | ✗ |
| ICEdit Zhang et al. (2025c) | 12B | Diffusion | ✗ |
| UltraEdit Zhao et al. (2024) | 1B | Diffusion | ✗ |
| AnyEdit Yu et al. (2025a) | 1B | Diffusion | ✗ |
| OmniGen Xiao et al. (2025a) | 3.8B | Diffusion | ✗ |
| Step1X-Edit Liu et al. (2025) | 19B | Diffusion | ✗ |
| Qwen-Image-Edit Wu et al. (2025a) | 20B | Diffusion | ✗ |
| Flux1-Kontext Batifol et al. (2025) | 12B | Diffusion | ✗ |
| Bagel w/o Think Deng et al. (2025) | 14B | Diffusion | ✗ |
| Bagel w/ Think Deng et al. (2025) | 14B | Diffusion | ✓ |
| VAREdit Mao et al. (2025) | 8.4B | AR | ✗ |
| DIM-Edit Zeng et al. (2025) | 4.6B | Diffusion | ✗ |

where $r_{\mathrm{match}} = \frac{|\mathcal{P}^*|}{\max(|\mathcal{L}|,|\mathcal{L}'|)}$ is the matching rate, $\bar{c}.$ denotes average consistency scores across matched pairs, and $\{\omega.\}$ are empirically set weights (0.25, 0.2, 0.2, 0.2, 0.15 respectively). This metric provides a comprehensive assessment of layout preservation that is robust to layer count variations and sensitive to both geometric distortions and structural changes.

## C    Experiments

### C.1    Baseline Models

**Baseline Open-source Models** We evaluate on 13 open-source models covering auto regressive and diffusion-based framework. The model size ranges from 1B to 20B. The details of each model are shown in Table 7.

### C.2    MiLDEScore

In Table 2, we set $\tau = 0.3$, $k = 10.0$, $w_{if} = 0.30$, $w_{lc} = 0.30$, $w_{tr} = 0.30$, $w_a = 0.10$, and $w_{sy} = 0.15$. We discuss the reason we chose these in Section D.2 and D.3.

## D    MiLDEdit

### D.1    Preliminary of GRPO Algorithm

Group Relative Policy Optimization (GRPO) (Shao et al., 2024) has been proved to be helpful for improving reasoning capabilities for LLM (Shao et al., 2024), Multi-modal understanding (Huang et al., 2025) and even image generation (Zhang et al., 2025b; Jiang et al., 2025). GRPO computes advantages from a group of responses. Given each question-anwer pair $(q, a)$, old policy $\pi_{\theta_{\mathrm{old}}}$ randomly samples $G$ responses, denoted as $\{o_i\}_{i=1}^G$. Each response $o_i$ is then fed into a reward model to obtain a reward $R_i$. Then, the advantage of the $i$-th response is obtained by normalizing the rewards of the group:

$$A_i = \frac{\mathcal{R} - \mathrm{mean}(\{\mathcal{R}_i\}_{i=1}^G)}{\mathrm{std}(\{\mathcal{R}_i\}_{i=1}^G)} \tag{8}$$

GRPO applies a clipped objective similar to PPO with a KL penalty term:

$$\mathcal{J}_{\text{GRPO}}(\theta) = \mathbb{E}_{(q,a) \sim \mathcal{D}, \{o_i\}_{i=1}^G \sim \pi_{\theta_{\text{old}}}(\cdot|q)} \left[ \frac{1}{\sum_{i=1}^G |o_i|} \right.$$

$$\left. \sum_{i=1}^G \sum_{t=1}^{|o_i|} \left( \min\left(r_{i,t}(\theta)\hat{A}_i, \text{clip}(r_{i,t}(\theta), 1-\varepsilon, 1+\varepsilon)\hat{A}_i\right) - \beta D_{\text{KL}}(\pi_\theta \,\|\, \pi_{\text{ref}}) \right) \right]$$

(9)

where $r_{i,t}(\theta)$ is the important weight for each token $t$:

$$r_{i,j}(\theta) = \frac{\pi_\theta\left(o_{i,j} \mid q, o_{i,<j}\right)}{\pi_{\theta_{\text{old}}}\left(o_{i,t} \mid q, o_{i,<j}\right)}. \tag{10}$$

Usually, in the reasoning task with only textual output, the model is asked to generate responses following a structured format. The total rewards consist of two rule-based rewards: (1) format reward and (2) the evaluation metrics of the specific downstream task.

### D.2 More Details in MiLDEScore

**Motivation and Rescaling.** The motivation to design the MiLDEScore is to find a comprehensive metric considering all aspects in design document editing that can overall assess the quality of the edited document. Another reason is that we should not consider each criteria separately. For example, one model may have very high layout consistency score but low instruction following score. This means that the model fails to edit the document or directly return the original document to users. In this way, a high layout consistency score is meaningless. In order to aggregate the four aspects together, we need to first scale them into the same scope. According to the Aesthetic model (aes), the scope is ranging from 1 to 10, while the other three aspects ranging from 1 to 100. Therefore, we rescale them into the same scope by dividing aesthetic score by 10 and dividing other three scores by 100.

**Other Baselines.** There exist multiple ways to aggregate the four metrics into an overall score. We compare our proposed method with four representative baselines: (1) **DW_sum (Direct Weighted Sum)**, (2) **GeoMean (Geometric Mean Aggregation)**, and (3) **HCoreSup (Harmonic Core–Support Aggregation)**. Each baseline captures different assumptions about metric interactions.

**(1) Direct Weighted Sum (DW_sum).** The most straightforward way is a linear weighted combination of the normalized scores:

$$S_{\text{DW}} = w_{if} \cdot IF_h + w_{tr} \cdot TR_h + w_{lc} \cdot LC_h + w_a \cdot A_h.$$

This method assumes each metric contributes independently and linearly. Although simple and smooth, it tends to overestimate models that exhibit high layout consistency but poor instruction following, failing to penalize unedited outputs.

**(2) Geometric Mean (GeoMean).** The geometric mean combines all criteria multiplicatively:

$$S_{\text{GM}} = \left( (IF_h)^{w_{if}} \cdot (TR_h)^{w_{tr}} \cdot (LC_h)^{w_{lc}} \cdot (A_h)^{w_a} \right)^{1/\sum w},$$

which enforces that any low-dimensional score (e.g., a very low $IF_h$) will significantly lower the final score. This method penalizes unbalanced models but may underestimate systems that excel in one dimension while being average in others, leading to overly conservative evaluation.

**(3) Harmonic Core–Support (HCoreSup).** We divide metrics into "core" (*instruction following, text rendering*) and "support" (*layout consistency, aesthetics*) groups:

$$S_{\text{HC}} = \frac{2 \cdot S_{\text{core}} \cdot S_{\text{sup}}}{S_{\text{core}} + S_{\text{sup}}}, \quad S_{\text{core}} = \frac{w_{if} \cdot IF_h + w_{tr} \cdot TR_h}{w_{if} + w_{tr}}, \quad S_{\text{sup}} = \frac{w_{lc} \cdot LC_h + w_a \cdot A_h}{w_{lc} + w_a}.$$

This harmonic mean encourages balanced performance between content correctness and visual consistency, while still allowing partial compensation between the two groups.

Table 8: Ablation study on weight parameters and comparison of different scoring functions. All configurations satisfy $w_{if} + w_{lc} + w_{tr} + w_a = 1$. For MiLDEScore, the synergy weight $w_{sy}$ is fixed at 0.15 unless otherwise noted. $\rho$ denotes Spearman correlation with human evaluation. DW_sum, GeoMean, and HCoreSup do not utilize the synergy term.

| Configuration | $w_{if}$ | $w_{lc}$ | $w_{tr}$ | $w_a$ | $w_{sy}$ | Spearman $\rho$ | | | |
| --- | --- | --- | --- | --- | --- | --- | --- | --- | --- |
| | | | | | | MiLDEScore | DW_sum | GeoMean | HCoreSup |
| **Ours (Optimal)** | **0.30** | **0.30** | **0.30** | **0.10** | **0.15** | **0.88** | 0.58 | 0.61 | 0.79 |
| *Varying Primary Weights* | | | | | | | | | |
| IF Dominant (High) | 0.45 | 0.25 | 0.20 | 0.10 | 0.15 | 0.82 | 0.61 | **0.64** | 0.80 |
| IF Dominant (Mid) | 0.40 | 0.25 | 0.25 | 0.10 | 0.15 | 0.85 | **0.62** | 0.63 | **0.81** |
| LC Dominant (High) | 0.25 | 0.45 | 0.20 | 0.10 | 0.15 | 0.81 | 0.43 | 0.48 | 0.70 |
| LC Dominant (Mid) | 0.25 | 0.40 | 0.25 | 0.10 | 0.15 | 0.84 | 0.47 | 0.51 | 0.72 |
| TR Dominant (High) | 0.20 | 0.25 | 0.45 | 0.10 | 0.15 | 0.79 | 0.49 | 0.53 | 0.71 |
| TR Dominant (Mid) | 0.25 | 0.25 | 0.40 | 0.10 | 0.15 | 0.83 | 0.52 | 0.55 | 0.74 |
| A Dominant | 0.25 | 0.25 | 0.25 | 0.25 | 0.15 | 0.76 | 0.41 | 0.46 | 0.65 |
| Equal Weights | 0.25 | 0.25 | 0.25 | 0.25 | – | 0.76 | 0.45 | 0.52 | 0.71 |
| *Varying Synergy Weight (MiLDEScore only)* | | | | | | | | | |
| No Synergy | 0.30 | 0.30 | 0.30 | 0.10 | 0.00 | 0.813 | – | – | – |
| Low Synergy | 0.30 | 0.30 | 0.30 | 0.10 | 0.05 | 0.842 | – | – | – |
| High Synergy | 0.30 | 0.30 | 0.30 | 0.10 | 0.25 | 0.856 | – | – | – |
| Very High Synergy | 0.30 | 0.30 | 0.30 | 0.10 | 0.30 | 0.831 | – | – | – |

As is shown in this table, our sigmoid-gated synergistic method achieves the highest consistency with human ratings, showing that incorporating soft gating and interaction terms better captures subjective quality assessment.

### D.3 Ablation Study on Weight Parameters

To validate the effectiveness of our proposed evaluation metric and determine the optimal weight configuration, we conduct comprehensive ablation studies on the weight parameters. Our evaluation score is formulated as:

$$\text{Score} = w_{if} \cdot \hat{IF} + w_{tr} \cdot \hat{TR} + g \cdot (w_{lc} \cdot \hat{LC} + w_a \cdot \hat{A}) + w_{sy} \cdot g \cdot \hat{IF} \cdot \hat{LC} \qquad (11)$$

where $\hat{IF}$, $\hat{LC}$, $\hat{TR}$, and $\hat{A}$ denote the normalized scores for Instruction Following, Local Consistency, Text Rendering, and Aesthetics, respectively. The gating function $g = \sigma(k(\hat{IF} - \tau))$ modulates the contribution of consistency and aesthetics based on instruction following performance, with $\tau = 0.3$ and $k = 10.0$. The synergy term $w_{sy} \cdot g \cdot \hat{IF} \cdot \hat{LC}$ captures the multiplicative interaction between instruction following and local consistency.

**Experimental Setup.** We systematically evaluate different weight configurations while satisfying the constraint $w_{if} + w_{lc} + w_{tr} + w_a = 1$. To assess the alignment between our automatic metric and human judgment, we compute the Spearman rank correlation coefficient ($\rho$) between the scores produced by each configuration and human evaluation scores collected from expert annotators.

**Results and Analysis.** As shown in Table 8, our optimal configuration ($w_{if} = 0.30$, $w_{lc} = 0.30$, $w_{tr} = 0.30$, $w_a = 0.10$, $w_{sy} = 0.15$) achieves the highest Spearman correlation of $\rho = 0.88$ with human evaluation, significantly outperforming alternative configurations. We analyze the impact of each design choice:

**(1) Balanced vs. Dominant Weights.** Configurations that heavily favor a single dimension (IF Dominant, LC Dominant, or TR Dominant with weights of 0.45) yield substantially lower correlations ($\rho = 0.650$), indicating that no single metric alone captures the multifaceted nature of image editing quality. Similarly, the Equal Weights configuration ($w_{if} = w_{lc} = w_{tr} = w_a = 0.25$) achieves only $\rho = 0.671$, suggesting that treating aesthetics equally with other dimensions does not align well with human preferences.

**(2) Role of the Synergy Term.** The synergy term proves crucial for capturing the interaction between instruction following and local consistency. Removing this term entirely (No Synergy, $w_{sy} = 0$) reduces the correlation to $\rho = 0.692$, while excessive synergy weighting (High Synergy, $w_{sy} = 0.30$) yields a similar degradation ($\rho = 0.692$). This demonstrates that moderate synergy ($w_{sy} = 0.15$) effectively models how human evaluators reward edits that simultaneously follow instructions accurately and maintain visual coherence.

**(3) Comparison with Alternative Scoring Functions.** We further compare MiLDEScore against three baseline aggregation methods: Direct Weighted Sum (DW_sum), Geometric Mean (GeoMean), and Harmonic Core-Support (HCoreSup). As shown in Table 8, MiLDEScore consistently outperforms all baselines across different weight configurations. Under the optimal setting, MiLDEScore achieves $\rho = 0.88$, substantially surpassing HCoreSup ($\rho = 0.79$), GeoMean ($\rho = 0.61$), and DW_sum ($\rho = 0.58$).

The performance gap stems from two key innovations in MiLDEScore: (i) the *adaptive gating mechanism* that dynamically modulates the contribution of visual quality metrics based on instruction following performance, preventing inflated scores for models that preserve content without executing the requested edit; and (ii) the *synergy term* that explicitly captures the positive interaction between instruction following and local consistency, which none of the baselines can model. Notably, even the best-performing baseline configurations (IF Dominant Mid for DW_sum and HCoreSup, IF Dominant High for GeoMean) achieve at most $\rho = 0.81$, still significantly below our optimal MiLDEScore. This consistent advantage across all configurations demonstrates that the architectural design of MiLDEScore, rather than parameter tuning alone, accounts for its superior alignment with human judgment.

One thing to mention is that we do not use the unedited layers before merging to measure the consistency. The reason is that we design this metric mainly for the one flat image editing scenario, where we cannot get layer images.

# E  Extended Analysis and Baseline Variations

To further validate the design choices of MiLDEdit and ensure a comprehensive and fair comparison with existing paradigms, we conduct two extended sets of experiments. First, we explore alternative multi-image input-output configurations for the baselines to evaluate whether implicitly providing layer information can achieve competitive results. Second, we investigate the impact of detailed prompt optimization and address the architectural constraints regarding the fine-tuning of image editors.

## E.1  Evaluation on Multi-Image Input Formats (MISO/MIMO)

An intuitive extension for flat image editing baselines to handle layered tasks is to provide individual layers as additional visual inputs. We formalize two potential settings for these baselines:

- **Multi-Input-Single-Output (MISO):** The model receives the flattened composite image alongside all individual layer images as inputs, and generates a single edited flattened image.

- **Multi-Input-Multi-Output (MIMO):** The model takes the same multi-image inputs but is tasked with directly outputting a complete set of edited layer images.

Among the open-source baselines evaluated, OmniGen (Xiao et al., 2025a) natively supports the MISO format. We also incorporate FLUX.2 (Labs, 2025). For the MIMO setting, since native layer-by-layer generation is primarily supported by advanced commercial systems, we evaluate GPT-Image-1 under both MISO and MIMO protocols.

As shown in Table 9, simply feeding individual layers as unstructured additional images does not yield the same benefits as explicit, structured layer-aware modeling. While GPT-Image-1 (MIMO) demonstrates strong capabilities in instruction following (23.9) and text rendering (25.1), its layout consistency drastically drops to 48.3. Similarly, MiLDEdit significantly outperforms OmniGen-MISO, FLUX.2-MISO, and GPT-Image-1-MISO in layout consistency (94.7 vs. 76.8, 57.6, and 59.2, respectively). These findings suggest that multi-image token concatenation alone is insufficient for the network to inherently deduce layer hierarchies,

precise editing targets, or complex compositional constraints. In contrast, MiLDEdit explicitly conduct layer-wise reasoning and editing, and successfully preserving the global layout layout during edits.

Table 9: Quantitative comparison under Multi-Input-Single-Output (MISO) and Multi-Input-Multi-Output (MIMO) baseline settings.

| Model | Instruction Following | Layout Consistency | Text Rendering |
|---|---|---|---|
| MiLDEdit (7B)-Flux | 17.9 | **94.7** | 23.5 |
| OmniGen (MISO) | 8.2 | 76.8 | 14.7 |
| FLUX.2 (MISO) | 14.7 | 57.6 | 19.5 |
| GPT-Image-1 (MISO) | 23.1 | 59.2 | 24.6 |
| GPT-Image-1 (MIMO) | **23.9** | 48.3 | **25.1** |

### E.2 Impact of Advanced Prompting and Fine-Tuning Constraints

We clarify that MiLDEdit does not fine-tune the core image editing backbone. Instead, GRPO is leveraged solely to train a text-based reasoning framework that maps user instructions and document structures to relevant target layers and explicit editing instructions. Directly applying reinforcement learning or fine-tuning flat image editors on this task presents severe dataset and pipeline bottlenecks: as is discussed in Sec. 3, our benchmark does not contain ground-truth edited documents.

To bridge this comparison and explore the limits of flat editing without editor fine-tuning, we introduce an **Oracle-Detailed Prompt Flat Editing** baseline. For each test case, we utilize the ground-truth layer-wise editing steps and employ GPT to synthesize them into an optimized, highly detailed global prompt for editing the flattened image. This baseline bypasses the target-layer inference bottleneck by inheriting oracle layer-level information. To present an optimistic upper bound, we sample three unique global prompts per case and report the *Best-of-Three* (Bo3) results.

The experimental results evaluated on 100 randomly sampled cases are summarized in Table 10. While oracle-detailed prompting substantially boosts flat editing performance with GPT-Image-1 (Bo3) achieving superior instruction following (26.7) and text rendering (27.6) over MiLDEdit, it fails to resolve the fundamental problem of layout degradation, which is consistent with results in Table 2. Specifically, the layout consistency of GPT-Image-1 (Bo3) and Flux-1-Edit (Bo3) remains critically low at 60.3 and 51.1, respectively, compared to 94.7 achieved by MiLDEdit. This indicates that the bottleneck of flat image editors is not merely prompt expressiveness; rather, modifying a flattened composite inherently corrupts un-targeted and irrelevant regions. MiLDEdit alleviates this by decoupling the editing process into localized layer modifications and subsequently re-composing the unchanged layers.

Table 10: Comparison with flat editing baselines enhanced by Oracle-Detailed Prompting under a Best-of-Three (Bo3) evaluation protocol.

| Model | Instruction Following | Layout Consistency | Text Rendering |
|---|---|---|---|
| MiLDEdit (7B)-Flux | 17.9 | **94.7** | 23.5 |
| GPT-Image-1 (Bo3) | **26.7** | 60.3 | **27.6** |
| Flux-1-Kontext (Bo3) | 13.2 | 51.1 | 19.1 |

## F Analysis of Evaluation Question Bias.

A potential concern in LLM-based evaluation pipelines is model-specific bias, where an evaluation model might favor question styles aligned with its own generation distributions. In our design, utilizing the instruction-aware model to generate evaluation questions is motivated by the necessity that questions must be strictly grounded within the specific editing instructions and localized layer modifications. Without conditioning on

these fine-grained edit constraints, a generator tends to produce generic visual questions detached from the actual modification.

To systematically quantify this potential bias, we conduct a validation study leveraging a subset of 100 expert human-written evaluation questions. We compute the mutual semantic similarity across four distinct reference sets—Human-written, GPT-generated, InternVL-generated, and Random baselines (the latter generated solely from global instructions without layer-level details)—using the `all-MiniLM-L6-v2` sentence embedding model.

As illustrated in the cross-source similarity matrix (Figure 8), Human, GPT, and InternVL-generated questions exhibit high mutual semantic affinity ($\geq 0.717$). In sharp contrast, the Random baseline yields drastically lower similarity scores ($\sim 0.42$). These empirical alignments suggest that InternVL-generated questions successfully capture the core semantic essence of human intent rather than introducing arbitrary, model-specific artifacts. While this tightly grounded generation strategy effectively preserves evaluative precision, we acknowledge that deploying entirely decoupled or fully human-in-the-loop evaluation frameworks remains an important direction for scaling up the benchmark.

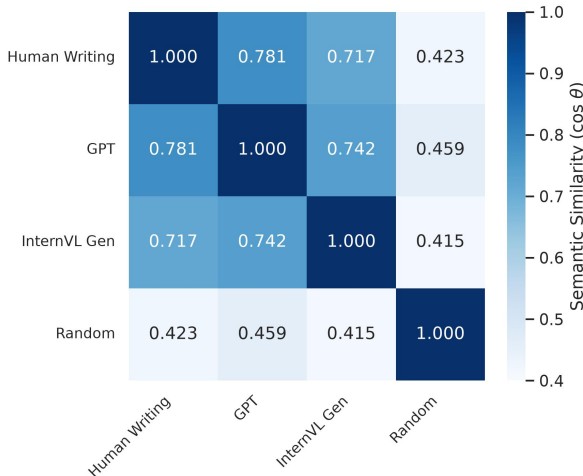

Figure 8: Semantic similarity matrix across different question generation sources.

## G  Failure Cases

**Failure Cases.** Our agent still exhibits certain failure modes. First, as layer decisions are made independently, multiple layers may occasionally be edited simultaneously, resulting in unintended overlaps or visual conflicts. Second, even with high-layer decision accuracy (e.g., the 7B model), The overall instruction-following score can remain low due to (i) ambiguous or underspecified layer-wise editing prompts, and (ii) the inherent limitations of the underlying image editing model. A potential solution is to integrate a self-checking mechanism that verifies the merged output and re-invokes editing when inconsistencies are detected. One example is shown in Figure 9. Our agent successfully predicts whether the layer should be edited. However, the merged document shows overlapped text and main image. This can be partially solved by self-checking mechanism in future. However, adding a self-checking mechanism is not the main story of our paper; therefore, we leave this part as our future plan.

## H  More Cases

We show more cases from Figure 9 to 11.

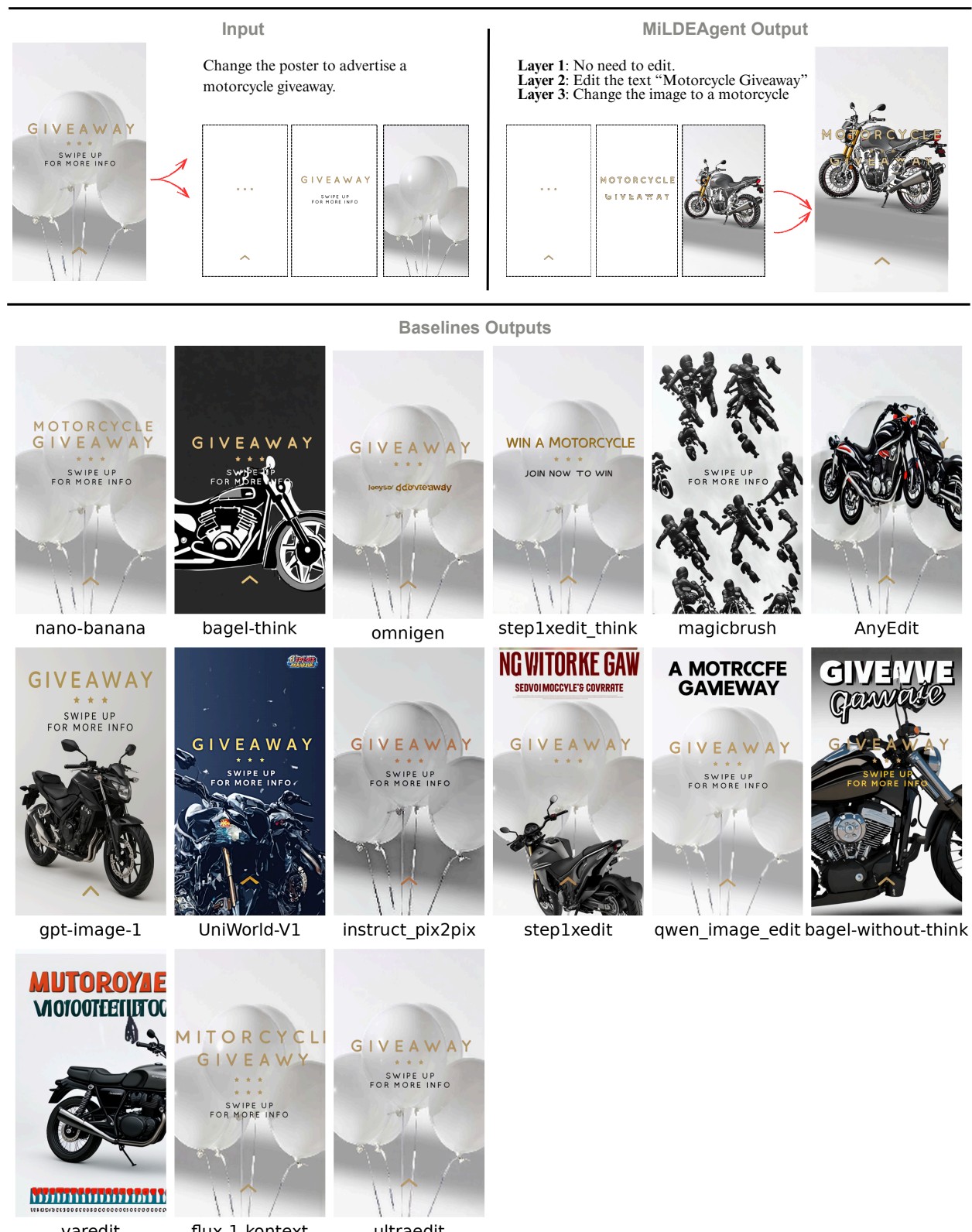

Figure 9: More examples 1.

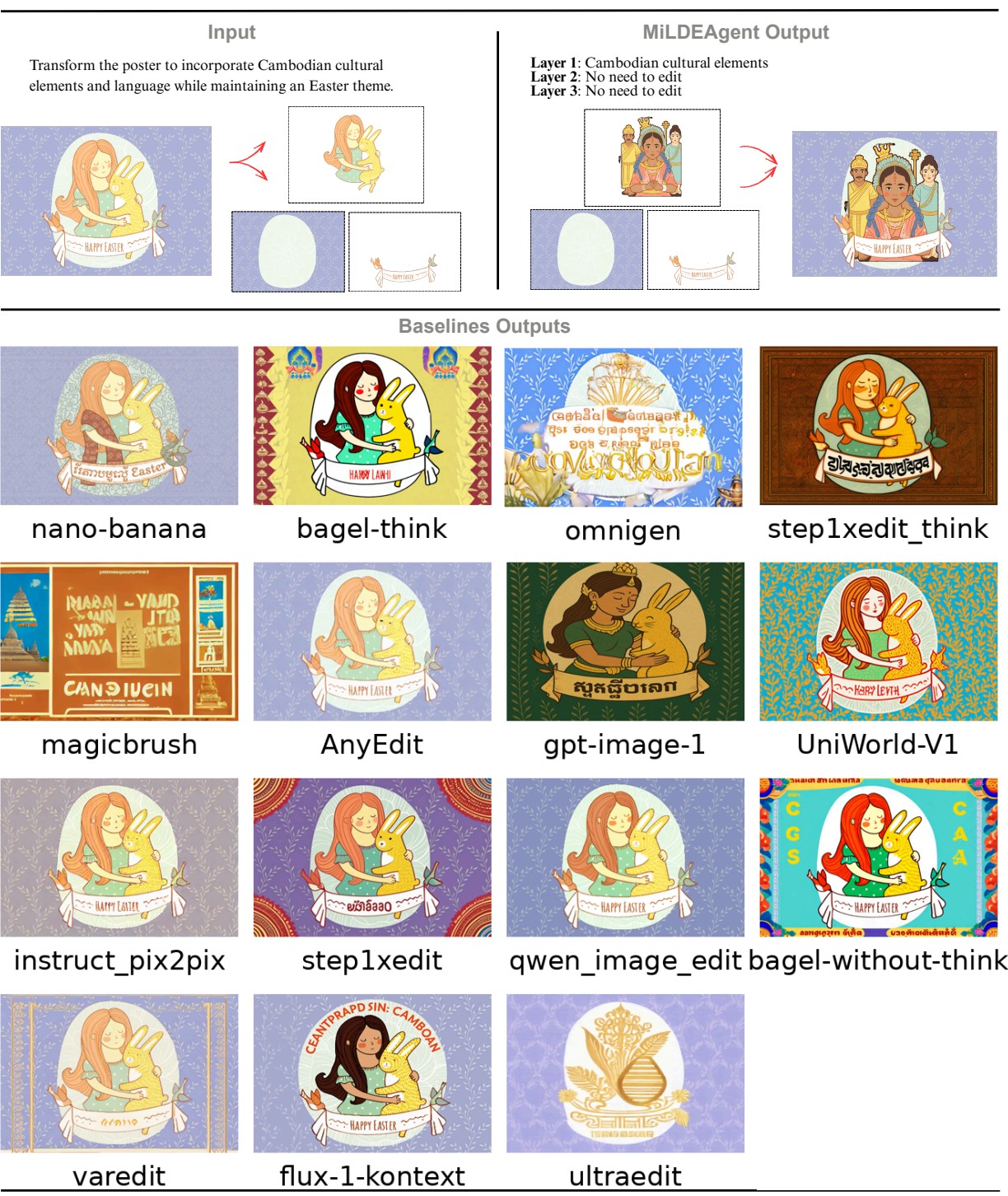

Figure 10: More examples 2.

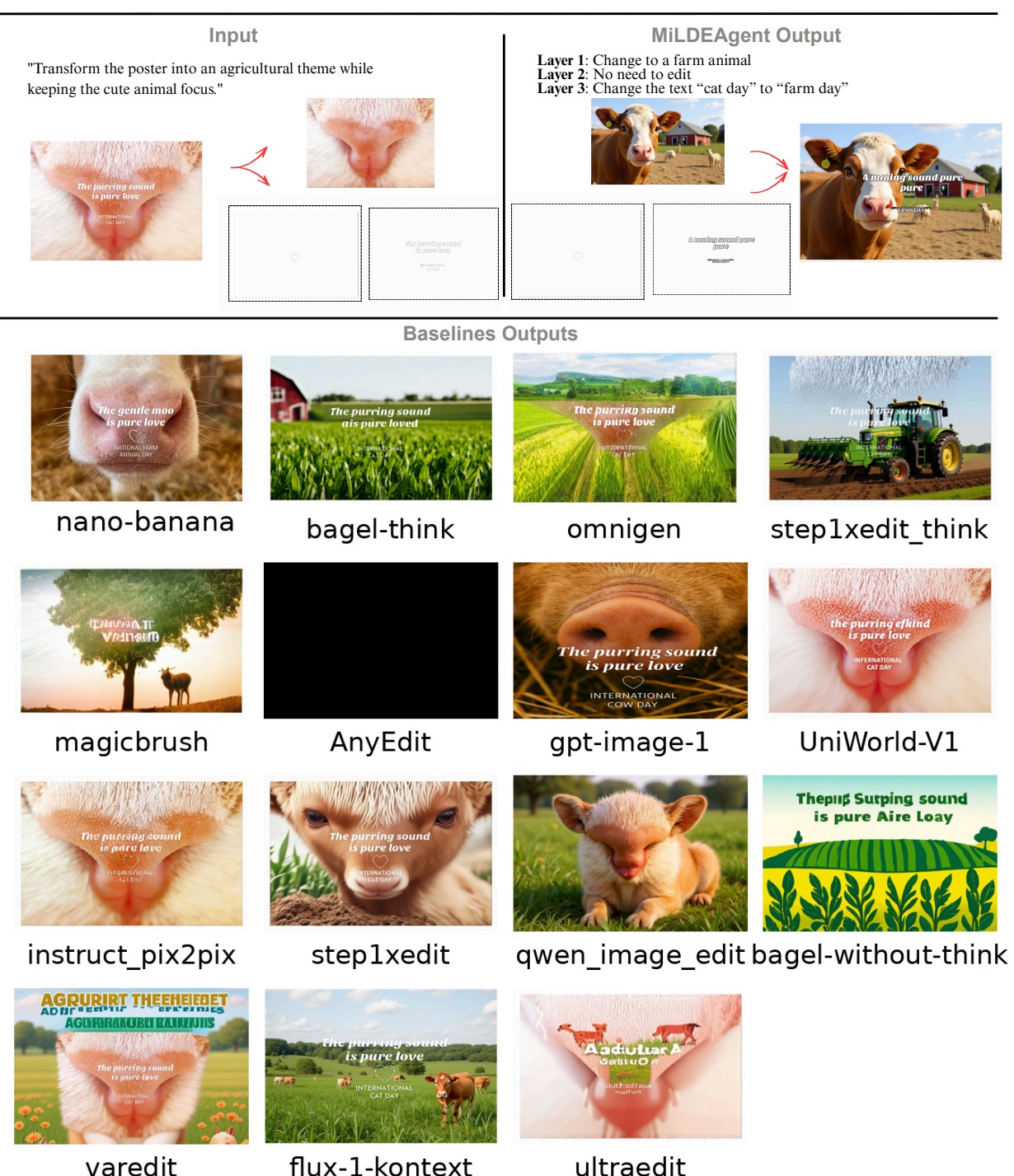

Figure 11: More examples 3.

