# OpenReview forum: "MiLDEdit: Reasoning-Based Multi-Layer Design Document Editing"
_TMLR — Under review for TMLR_

### Review · Reviewer_Fqp6 · 2026-04-17

**Summary Of Contributions:**

This paper points out that real design documents are multi-layered, with text, decoration, and images, so editing needs layer-aware reasoning instead of standard flat-image editing.
This paper makes three main contributions. It introduces MiLDEBench, with layered design documents, document-level instructions, relevant-layer annotations, and layer-wise edit instructions. It proposes MiLDEEval and MiLDEScore for instruction following, layout consistency, aesthetics, and text rendering, and presents MiLDEAgent, which combines a GRPO-trained reasoner with a frozen image editor for layer-wise editing and recomposition.

Strengths:

1. The task is meaningful. It focuses on a real difficulty in design documents, where multiple layers, text, and layout need to be handled together.

2. This benchmark is carefully constructed and the evaluation protocol is well designed for the task. It evaluates instruction following, layout consistency, aesthetics, and text rendering, and also proposes a unified score to reflect their interaction.

3. The empirical study is informative and convincing. The experiments show that current models perform poorly in this setting, and the proposed method gives clear gains over open-source baselines, especially on layout consistency, text rendering, and layer decision accuracy.

Weaknesses:

1. The proposed benchmark assumes access to layer decomposition. In this case, the setting is closer to structured source-file editing than to common raster poster editing. The paper should state this scope more directly when describing the task.

2. The task setting is still narrower than many practical design editing workflows. The paper disallows geometry changes such as text reshaping or object movement, but these operations are common in real design editing.

3. The claims should be interpreted within the task setting. The experiments support the method on layered transparent documents without geometry changes. Therefore, the conclusions are strongest for this specific structured editing setting, rather than for real-world design editing in a broader sense.

**Audience:**

Yes

**Audience Explanation:**

I believe part of the TMLR audience will be interested in this paper because it connects multimodal reasoning, image editing, and structured document understanding. The paper defines a new benchmark and task setting that exposes a clear weakness of current editing models. I also think the findings are interesting because the paper shows that simply adding thinking to current image editing models gives only limited gains in this structured editing setting. This is a useful finding for researchers working on reasoning-based generation and multimodal agents.

**Broader Impact Concerns:**

I do not see a major unaddressed ethical risk that is unique to this paper.

**Claims And Evidence:**

Yes

**Claims Explanation:**

The main claims are mostly supported by the experiments. Table 2 shows that existing open-source and closed-source baselines perform poorly on this benchmark, especially on instruction following and text rendering, and MiLDEAgent is clearly stronger than open-source baselines on the proposed metrics.

The claim that explicit reasoning training is useful is also supported. The ablation shows that the GRPO-trained reasoner improves layer decision accuracy strongly over zero-shot reasoners, and this matches the final editing results.

**Requested Changes:**

1. The paper should clarify earlier and more directly what real-world setting it covers and what setting it does not cover. The current task assumes layered transparent documents and disallows geometry changes.

2. The paper should make the limitation of independent layer decisions more visible in the main text. The current failure analysis shows that independent edits can still create overlap or conflict after merging.

3. The paper should improve writing clarity in several places. There are spelling issues and repeated expressions. For example, Section 4.1 repeats the sentence "We evaluate the faithfulness of edited text with an OCR–VQA pipeline" twice in a row. There are also small wording issues such as "multi-layer editng" on Page 7 and "focuing" on Page 2, which should be corrected.

---

> ### Author Response · Authors · 2026-06-27
> **Official Response to Reviewer Fqp6**
>
> We sincerely thank the reviewer for the constructive comments and for recognizing the value of our task, benchmark, evaluation protocol, and empirical findings. We address the reviewer’s concerns below.
>
> ## Requested Change 1: Clarifying the task scope, layer access assumption, and geometry-changing edits.
>
> We agree with the reviewer that the scope of our task should be stated more explicitly. Our work focuses on content-level editing of multi-layer design documents, where the editable layer structure is available as part of the document representation. This setting is closer to editing structured design source files, such as PSD-, Figma-, Canva-, or Crello-style documents, rather than editing arbitrary rasterized posters. We have revised the introduction and task formulation to clarify that our benchmark assumes access to layered transparent documents and that our conclusions should be interpreted within this structured editing setting.
>
> We also agree that geometry-changing edits, such as moving objects, reshaping text, resizing elements, or swapping the positions of text and image layers, are important in practical design workflows. Our current benchmark focuses on content-level layer editing while preserving the original layer geometry. To avoid overclaiming, we now explicitly state this scope in the main paper and discuss geometry-changing multi-layer editing as an important future direction in the new Limitations and Future Work section (Sec. 7) as the first main limitation of our work.
> To further examine the layer-access assumption, we added a small-scale ablation study using a layer decomposition model, e.g., Qwen-Image-Layered, to obtain pseudo-layers from a flattened poster image before applying our framework. The details of this experiment have been discussed in the rebuttal reply to Reviewer FyoP, the weakness 2. The results suggest that incorporating predicted layers is feasible and that our framework can still outperform training-free zero-shot baselines under this more realistic setting. We also added this experiment in Sec. 6.1 in our revised version.
>
> ## Requested Change 2: Making independent layer-decision limitations more visible in the main text.
>
> We agree that independent layer decisions can lead to conflicts after recomposition. As shown in our failure analysis, editing one layer may affect the visual compatibility of other layers. In the revised manuscript, we added a new Limitations and Future Work section (Sec. 7) before the conclusion to make this limitation explicit in the main text. We specify this independent layer decision limitation as the second main limitation of our work.
>
> ## Requested Change 3: Improving writing clarity and correcting typos.
>
> We thank the reviewer for carefully pointing out these writing issues. We have revised the manuscript to improve clarity and fix the identified errors. Specifically, we removed the duplicated sentence in Section 4.1, corrected “multi-layer editng” to “multi-layer editing,” and corrected “focuing” to “focusing.” We also performed an additional proofreading pass to fix other minor spelling and wording issues throughout the paper.

---

> > ### Comment · Reviewer_Fqp6 · 2026-06-30
> >
> > Thank you for the detailed rebuttal and revision. These additions address most of my concerns.
> >
> > My remaining comment is mainly about the boundary between transparent raster-layer editing and structured source-file editing. Since the main setting assumes access to layered documents, some metadata such as masks, bounding boxes, z-order, coarse layer types, and text content via OCR may be available or recoverable. It would be helpful to discuss whether a structured-operation baseline is possible, where an LLM/VLM planner predicts edit actions, directly edits text or simple design elements programmatically, then renders and verifies the result. If this is outside the paper’s intended scope, I suggest explicitly stating that the method operates on raster transparent layers rather than full editable source-file objects.
> >
> > I also think the claims about flat/raster poster editing should remain cautious, since the automatic decomposition experiment is small-scale and sensitive to layer quality.
> >
> > Overall, I am satisfied with the rebuttal, and my remaining suggestions are mainly about clarifying the task boundary and positioning.

---

> ### Comment · Action_Editor_gXXp · 2026-07-12
> **Official recommendation**
>
> Dear Reviewer Fqp6,
>
> Could you please submit your official recommendation?
>
> Best,
>
> AC

---

### Review · Reviewer_FyoP · 2026-06-03

**Summary Of Contributions:**

This paper considers the problem of multi-layer design document editing, which is a subtask of image editing. The paper argues that prior methods assume the image lies on a flat canvas, where reasoning is limited to where and how to edit. In contrast, this work focuses on the decomposed version of images. The authors formalize this problem into MiLDEBench, a dataset of documents paired with editing instructions, along with an evaluation protocol. They also propose a pipeline to reason about which layers to modify and how (layer-specific reasoning), fine-tuned with GRPO.


**Strengths:**

1. The problem formulation is clean and complete.

2. The writing is clear and coherent throughout.

3. Both the evaluation protocol and the composite score are novel and well-designed, serving the needs of both the GRPO training and inference-time assessment.


**Weaknesses:**

1. The word "agent" in the title is misleading, as this is a single-pass pipeline with no iterative refinement in the framework. Iterative editing is mentioned in the related work as one of the gaps MiLDEBench aims to address, yet there is no notion of iteration anywhere in the proposed method.

2. The assumption that a layered version of the image is always available is very strong. In practice, real-world layer decompositions may not be perfect. It would have been more appropriate to incorporate a layer decomposition model such as Qwen-Image-Layered into the framework to account for this limitation.

3. Most modern vision-language models support multiple images as input, But did the authors try providing the individual layers as additional input to the baselines? Furthermore, as I understand it, the proposed reasoner is fine-tuned specifically for layer-aware prompt generation, while the baselines receive only a plain instruction and must rely on their general-purpose internal representations to enrich it. A fairer comparison would involve GRPO fine-tuning the same baselines for flat image editing (not layer editing) and using them only as editors, to truly isolate the contribution of layer-aware reasoning.

4. The layout consistency score is questionable. Although the paper introduces a reasonable gating mechanism, would it not have been simpler and more reliable to compute consistency directly over the unedited($y_i = 0$) layers before merging?

5. In the instruction following metric, the same model that generates the questions is also the one answering them. This likely introduces bias toward the types of questions it tends to ask.

6. The reasoner is trained in an editor-agnostic manner, yet different editors have their own specific architectures and prompting requirements. Ablation 2 does not add much insight under this setting and it would be far more informative once editor-dependent training is introduced and compared.

**Audience:**

Yes

**Audience Explanation:**

Yes. The problem of multi-layer design document editing is new and interesting in practice, and many readers in TMLR's audience will find it relevant. The benchmark and evaluation protocol are well-designed and are good contributions by themselves. The findings about how  that adding reasoning helps and that bigger models do not always do better are interesting and worth knowing. With some fixes, this paper would be a useful addition to the field.

**Broader Impact Concerns:**

The main potential misuse of this work is editing documents for deceptive purposes, such as forging posters or manipulating visual content. This risk is not unique to this paper and applies to image editing research in general. A short broader impact statement acknowledging this would be sufficient.

**Claims And Evidence:**

Yes

**Claims Explanation:**

The claims are mostly supported. One notable exception is the use of the term "agent".
The proposed framework is a single-pass pipeline with no iterative refinement, feedback loops, or dynamic decision making, which does not align with the standard definition of an agent in the literature.

**Requested Changes:**

1. The method is not really an agent. It runs once and stops, with no re-checking or fixing of mistakes. The authors should use a more accurate term. This is especially odd because the paper itself mentions iterative editing as an important missing capability in prior work, yet does not include it.

2. The method requires the original layered file, which is often not available in practice. The authors should test what happens when layers are obtained from a decomposition model like Qwen-Image-Layered instead, to show the method works in more realistic settings.

3. Most modern vision models can take multiple images as input, did the authors try giving the baselines the individual layers too? Also, the proposed reasoner was specifically trained for this task while baselines use general-purpose models with no task-specific training. To fairly compare, the authors should fine-tune the same baselines on flat editing and use them only as editors, so the comparison isolates the value of layer-aware reasoning specifically.

4. Instead of the gating mechanism, why not just measure layout consistency on the layers that were not supposed to be edited ($y_i = 0$)? This would be more direct and easier to interpret.

5. The same model writes the evaluation questions and answers them. This is likely biased. The authors should at least discuss this limitation or validate the metric with human-written questions.

6. The reasoner is trained without knowing which editor will be used, but Ablation 2 shows the editor choice matters a lot. This means the reasoner is generating prompts that may not work well for specific editors. The authors should train with the editor in the loop, and then redo Ablation 2 to see if it tells a more useful story.

---

> ### Author Response · Authors · 2026-06-27
> **Official Response to Reviewer FyoP (1)**
>
> We sincerely thank Reviewer FyoP for the careful reading and constructive suggestions. We appreciate that the reviewer identified several important issues regarding terminology, layer availability, evaluation design, and editor dependence. We prove the response in the following paragraphs.
>
> ## Weakness 1 & Requested Change 1: Terminology of ``agent''.
>
> We thank the reviewer for pointing this out. We agree that the term “agent'' may imply an interactive decision-making process with iterative environment feedback, while our main framework is a single-pass reasoning-and-editing pipeline. To avoid ambiguity, in our revised version, we have changed the terminology throughout the paper by replacing “MiLDEagent'' with a more accurate description, namely a multi-layer document editing framework, in short “MiLDEdit”. Besides, lack of the iterative refinement process leads to one limitation that our current framework fails to solve the inter-layer conflicts as discussed in the rebuttal for weakness 2 for reviewer qctc. We have added one section “limitations and future work” to demonstrate this limitation (the second one) in the revised manuscript.
>
> ## Weakness 2 & Requested Change 2: Requirement of accessible transparent layers.
>
> We thank the reviewer for raising this important point. We agree that assuming access to perfect original layers can be restrictive in some real-world scenarios. Our main setting is motivated by a practical class of document-editing workflows where layer information is naturally available (e.g., Canva, Adobe Express, Figma, and PowerPoint), where users manipulate documents through editable text, image, shape, and decorative layers. Our goal is to study this layer-aware document editing setting, which is common in design tools but has not been systematically formulated and evaluated before.
>
> However, to address the scenario where models do not have access to layer information, following the reviewer's suggestion, we applied Qwen-Image-Layered (QIL) as an automatic decomposition module. Specifically, we randomly sampled 100 examples, decomposed each flat document into multiple layers using QIL, and then fed them into our framework without any retraining or architectural modification, using Flux-1-Kontext as the image editor. Since the decomposition model requires a hyperparameter to specify the output layer number, we evaluated three configurations: the default 4-layer setting, an over-decomposition setting with 10 layers, and an oracle setting given the ground-truth number of layers (accessible in MiLDEBench).
>
> As shown in the table below, using QIL, the default 4-layer setting achieves 16.8 instruction following, 92.9 layout consistency, and 20.7 text rendering, while the oracle setting further improves to 17.1, 93.4, and 21.6, closely approaching the performance with original layers (17.9, 94.7, 23.5). In contrast, the 10-layer setting drops significantly to 11.2, 87.5, and 16.8. This drop occurs because over-decomposition often produces redundant or duplicated layers, such as splitting a single text element into several visually similar layers. Since MiLDEBench contains 4.4 layers on average, the 4-layer and oracle settings provide more accurate decompositions and preserve most of the benefits of layer-aware editing. These results demonstrate that our framework remains highly effective when layers are automatically recovered, and its performance scales with better layer decomposition quality. We have added this experiment to Section 6.1 (*Extension to Automatically Decomposed Layers*) of our revised manuscript.
>
> | Model | Instruction Following | Layout Consistency | Text Rendering |
> | :--- | :---: | :---: | :---: |
> | **MiLDE (7B) - Flux (Original Layers)** | **17.9** | **94.7** | **23.5** |
> | + QIL (4 layers) | 16.8 | 92.9 | 20.7 |
> | + QIL (10 layers) | 11.2 | 87.5 | 16.8 |
> | + QIL (Oracle) | 17.1 | 93.4 | 21.6 |

---

> ### Author Response · Authors · 2026-06-27
> **Official Response to Reviewer FyoP (2)**
>
> ## Weakness 3 & Requested Change 3: Fairness of comparison and layer inputs for baselines.
>
> ### Experiment 1: multi image input and edit multi image simultaneously
>
> We thank the reviewer for this valuable suggestion. We agree that providing individual layers as additional inputs is an important comparison. In our paper, however, this setting was not adopted because most baselines in Table 1 do not support such input-output formats, except OmniGen. However, we acknowledge that closed-source models, e.g., GPT-Image-1 can take multiple images as input and edit them simultaneously. Besides, we found that FLUX.2 model can also do that.
>
> Following the reviewer’s suggestion, we add the comparison experiments and consider two separate settings: (1) multi-input-single-output (MISO), where the model takes the flattened image and all individual layer images as input and outputs one edited flattened image; and (2) multi-input-multi-output (MIMO), where the model takes the same inputs but outputs a set of edited layer images. Among the original baselines, only OmniGen supports MISO. We also include FLUX.2 for MISO. For MIMO, we found that this capability is currently mainly supported by closed-source models, so we evaluate GPT-Image-1 under both MISO and MIMO.
>
> The results show that simply feeding layers as additional images does not provide the same benefits as explicit layer-aware modeling. For example, MiLDE achieves much higher layout consistency than OmniGen-MISO (94.7 vs. 76.8), FLUX.2-MISO (94.7 vs. 57.6), and GPT-Image-1-MISO (94.7 vs. 59.2). Although GPT-Image-1-MIMO achieves strong instruction following and text rendering scores (23.9 and 25.1), its layout consistency drops to 48.3. These results suggest that multi-image input alone does not sufficiently encode layer hierarchy, editing targets, or compositional constraints. We have added this ablation study in the Appendix E.1 in our revised manuscript.
>
> | Model | Instruction Following | Layout Consistency | Text Rendering |
> | :--- | :---: | :---: | :---: |
> | MiLDE (7B)-Flux | 17.9 | 94.7 | 23.5 |
> | OmniGen (MISO) | 8.2 | 76.8 | 14.7 |
> | FLUX.2 (MISO) | 14.7 | 57.6 | 19.5 |
> | GPT-Image-1 (MISO) | 23.1 | 59.2 | 24.6 |
> | GPT-Image-1 (MIMO) | 23.9 | 48.3 | 25.1 |
>
> ### Experiment 2: isolates the value of layer-aware reasoning
>
> We thank the reviewer for raising this important point regarding baseline fairness. We clarify that in our framework, GRPO is exclusively utilized to train the text-based reasoner, which maps a "general global instruction" into "layer-wise editing prompts" anchored to specific layer images. The reviewer's comment points to two alternative baseline paradigms, which we address below:
>
> **(1) Directly Fine-tuning the Image Editor (Alternative 2):** Applying GRPO directly to fine-tune an image editor for flat-image editing would require paired pre-/post-edit raster images or a reliable image-level RL reward function. However, as detailed in Sec. 3.2, MiLDEBench does *not* human-validated ground-truth edited raster images. Consequently, optimizing a reliable image editor via RL under this setting is currently infeasible. We acknowledge this as a limitation and have discussed it in Sec. 7 of the revised manuscript.
>
> **(2) Reasoning via Detailed Prompts on Flattened Images (Alternative 1):** To rigorously evaluate this setting, we introduced an **Oracle-Detailed Prompt Flat Editing** baseline. For each test case, we synthesized the ground-truth layer-wise editing steps via GPT into a comprehensive, detailed global prompt tailored for the flattened single image. This baseline serves as an optimistic upper bound for prompt-based flat editing, bypassing the need to infer target layers. To maximize its potential, we sampled three prompts per case and reported the Best-of-Three (Bo3) performance. We evaluate this experiments on 100 randomly selected samples.
>
> As shown in the table below, while oracle-detailed prompting improves instruction following and text rendering for flat-image editors (e.g., GPT-Image-1 (Bo3) achieves 26.7 and 27.6, respectively), it fails to address the core challenge of layout preservation. GPT-Image-1 (Bo3) and Flux-1-Kontext (Bo3) both suffer from drastically lower layout consistency (60.3 and 51.1) compared to MiLDE (94.7). This empirically demonstrates that the bottleneck of flat-image editing is not merely prompt descriptiveness; rather, editing a flattened canvas inherently corrupts unrelated regions. In contrast, MiLDEdit natively preserves document layout by isolating edits to target layers and accurately recomposing unchanged elements. We have incorporated this experiment into Appendix E.2 of our revised manuscript.
>
> | Model | Instruction Following | Layout Consistency | Text Rendering |
> | :--- | :---: | :---: | :---: |
> | **MiLDE (7B)-Flux (Ours)** | **17.9** | **94.7** | **23.5** |
> | GPT-Image-1 (Bo3) | 26.7 | 60.3 | 27.6 |
> | Flux-1-Kontext (Bo3) | 13.2 | 51.1 | 19.1 |

---

> ### Author Response · Authors · 2026-06-27
> **Official Response to Reviewer FyoP (3)**
>
> ## Weakness 4 & Requested Change 4: Layout consistency metric.
>
> We thank the reviewer for the suggestion. We agree that, if all methods produced editable layers, directly measuring consistency over the unedited layers would be a simple and intuitive metric. However, this is not feasible in our benchmark setting because most existing baselines take a single composed image as input and output a single composed image. They do not produce a layer decomposition, nor do they indicate which regions or layers are intended to remain unchanged. Therefore, directly computing consistency on unedited layers would only be applicable to our method, but not to flat-image editing baselines, making the metric unsuitable for fair comparison across methods.
>
> Although one could apply an external layer decomposition model to the outputs of all baselines, this would introduce additional noise and ambiguity as we discussed in Weakness 2. In particular, automatically decomposed layers may not align with the original semantic layers, and it is difficult to reliably determine which decomposed component corresponds to an unedited layer. For this reason, we proposed the gated layout consistency score, which is designed to evaluate layout preservation in a way that can be consistently applied to both flat-image and layer-aware methods.
> We have added one description about this choice for the layout consistency metric in the end of Appendix D.3. in our revised version.
>
> ## Weakness 5 & Requested Change 5: Potential bias in instruction-following evaluation.
>
> We thank the reviewer for highlighting this important methodological consideration. Our evaluation pipeline intentionally conditions the question-generation model on both the specific layer-wise edit instruction and the corresponding edited layer. This design ensures that the generated evaluation questions are tightly grounded in the precise modifications, avoiding generic visual questions that fail to reflect the intended edit.
>
> To rigorously address potential model biases as noted by the reviewer, we conducted a validation study using 100 expert human-written evaluation questions as a benchmark. We computed the semantic similarity using `all-MiniLM-L6-v2` across four distinct sets: (1) Human-written, (2) GPT-generated, (3) InternVL3-38B-generated (our default), and (4) Ablated questions (generated by InternVL3-38B given *only* the global instruction, without layer-level editing context).
>
> As shown in the similarity matrix below, InternVL-generated questions demonstrate high semantic alignment with both human-written (0.717) and GPT-generated (0.742) questions, closely mirroring the human-GPT baseline similarity (0.781). Conversely, the ablated questions show a sharp drop in similarity across all benchmarks (averaging ~0.43). This confirms that InternVL-generated questions capture meaningful, human-aligned edit details rather than producing arbitrary artifacts or model-specific biases. We agree that incorporating fully human-verified evaluations or an independent evaluator model is a valuable trajectory, and we have added this comprehensive analysis to Appendix F of the revised manuscript.
>
> | Metric (Sentence Similarity) | Human-Written | GPT-Generated | InternVL-Generated | Ablated (w/o layer context) |
> | :--- | :---: | :---: | :---: | :---: |
> | **Human-Written** | 1.000 | 0.781 | 0.717 | 0.423 |
> | **GPT-Generated** | 0.781 | 1.000 | 0.742 | 0.459 |
> | **InternVL-Generated** | 0.717 | 0.742 | 1.000 | 0.415 |
> | **Ablated (w/o layer context)** | 0.423 | 0.459 | 0.415 | 1.000 |

---

> ### Author Response · Authors · 2026-06-27
> **Official Response to Reviewer FyoP (4)**
>
> ## Weakness 6 & Requested Change 6: Editor-agnostic reasoner and editor-dependent prompting.
>
> We thank the reviewer for this insightful suggestion. Our original framework intentionaly decoupled the text reasoner from specific downstream image editors (**editor-agnostic**). This design stems from the fact that different image editors exhibit distinct architectures, prompting preferences, and instruction-following behaviors. Our primary objective is to develop a general-purpose reasoning module that seamlessly pairs with various plug-and-play editors, rather than overfitting to a single model.
>
> Nevertheless, we agree that co-optimizing the reasoner with a specific target editor is a compelling direction. To explore this, we implemented an **editor-in-the-loop** experiment targeting Flux-1-Kontext. Since direct RL fine-tuning with full diffusion denoising rollouts is computationally prohibitive on the fly, we adopted a *rejection sampling* strategy. For each training instance, we sampled 5 candidate prompts from InternVL, executed the editing using Flux-1-Kontext, and utilized Qwen2.5-VL (72B) to select the prompt that best satisfied the target instruction. We then fine-tuned the reasoner from scratch using these Flux-optimized prompts as targets.
>
> We evaluated this **MiLDE-Flux-Specific** variant on the full test set across different downstream editors. As shown below, optimizing for Flux improves its native MiLDEScore from 25.94 to 26.73, with clear gains in instruction following and text rendering. However, when this same Flux-specific reasoner is transferred to Qwen-Image-Edit, performance drops from 26.73 to 23.50 (especially in instruction following). These results substantiate a clear trade-off: editor-in-the-loop training successfully aligns the reasoner with a specific target editor but compromises cross-editor transferability. We have incorporated this full analysis and table into Section 6.2 of our revised manuscript.
>
> | Reasoner Variant | Downstream Editor | Instruction Following | Layout Consistency | Aesthetic | Text Rendering | MiLDEScore |
> | :--- | :--- | :---: | :---: | :---: | :---: | :---: |
> | **Original (Editor-Agnostic)** | Flux-1-Kontext | 20.71 | 93.24 | 4.19 | 36.75 | 25.94 |
> | **Original (Editor-Agnostic)** | Qwen-Image-Edit | 19.12 | 92.86 | 4.21 | 34.25 | 23.92 |
> | **MiLDE-Flux-Specific** | Flux-1-Kontext | **22.64** | 93.15 | **4.31** | **39.65** | **26.73** |
> | **MiLDE-Flux-Specific** | Qwen-Image-Edit | 17.30 | 91.31 | 4.26 | 35.88 | 23.50 |

---

### Review · Reviewer_qctc · 2026-06-14

**Summary Of Contributions:**

This paper addresses the novel and challenging task of multi-layer design document editing (e.g., posters, flyers) from natural language instructions. Unlike prior work that treats images as flat canvases, the authors argue that real-world design documents have a layered, hierarchical structure. Their main contributions are threefold:
1. A New Benchmark (MiLDEBench): The first large-scale benchmark for reasoning-based, multi-layer document editing. It contains over 20K design documents with 50K editing instructions and 87K layer-aligned steps, created via a hybrid pipeline that combines multimodal LLMs with human-in-the-loop validation.
2. A Task-Specific Evaluation Protocol (MiLDEEval): A multi-dimensional metric (MiLDEScore) that evaluates edits across four dimensions: instruction following, layout consistency, aesthetics, and text rendering. The metric uses a sigmoid-gated synergistic formulation that correlates well with human judgment.
3. A new Agent (MiLDEAgent): A two-stage framework consisting of an RL-trained (GRPO) multimodal reasoner for layer identification and prompt synthesis, paired with a frozen image editor for targeted modifications. This agent significantly outperforms open-source baselines and approaches closed-source model performance while maintaining superior layout consistency.

Key Strengths:
* The paper identifies a real-world gap in document editing.
* High-quality benchmark: The human-in-the-loop and persona-based instruction generation add realism.
* Strong empirical results: The GRPO-trained reasoner shows a clear 4× improvement over zero-shot in layer decision accuracy.

Key Weaknesses (to be mindful of, see also Q4):
* The paper claims a human correlation of 0.908 for MiLDEScore in Appendix D.3 but reports 0.88 in Table 6, indicating a typo.
* The framework treats each layer independently, overlooking inter-layer dependencies; consequently, edits to one layer (e.g., replacing a background) may render another layer's content unreadable (e.g., text lacking sufficient contrast), revealing a need for future integration of coordinated foreground & background contrast reasoning to achieve visually harmonious outcomes.

**Additional Comments:**

please see comments above

**Audience:**

Yes

**Audience Explanation:**

* Researchers in multimodal LLMs and vision-language reasoning will find the GRPO-trained layer-aware reasoner a compelling application.
* Users of document AI and creative design tools will see direct practical value.
* Those working on evaluation metrics for generative models will find the MiLDEScore design (gated synergy term) a valuable contribution.
* The finding that shallow textual planning is insufficient and that deep multimodal reasoning integration is required is a generalizable insight beyond document editing. It speaks to a core challenge in aligning language instructions with structured visual outputs.

**Broader Impact Concerns:**

No ethical issues

**Claims And Evidence:**

Yes

**Claims Explanation:**

Claim 1 (Existing models struggle): Supported by Table 2, which benchmarks 15 models. Open-source models achieve only ~10% instruction following, and closed-source models (GPT-Image-1, Nano Banana) trade layout consistency for semantic alignment. This is clear and convincing.

Claim 2 (MiLDEAgent outperforms baselines): Strongly supported. MiLDEAgent achieves a more than 80% improvement in MiLDEScore over the best open-source baseline.

Claim 3 (Reasoning matters, but shallow reasoning is insufficient): The paper shows that simply adding "thinking" to models (Step1X-Edit w/ Thinking, Bagel w/ Thinking) yields only marginal gains. In contrast, the GRPO-trained reasoner in MiLDEAgent achieves substantial improvement. This supports the claim that deep multimodal reasoning integration is required, not just shallow textual planning. This is a particularly strong and insightful finding.

Claim 4 (MiLDEScore aligns with human judgment): The ablation in Appendix D.3 (Table 6) shows MiLDEScore achieves a Spearman correlation of 0.88 with human evaluators, outperforming alternatives (DW_sum: 0.58, GeoMean: 0.61, HCoreSup: 0.79). However, there is an internal inconsistency: the text in Appendix D.3 states the correlation is 0.908, while Table 6 shows 0.88. This needs correction.

Failure cases acknowledged: The paper transparently discusses failure modes (e.g., overlapping edits, limitations of the underlying image editor), which strengthens credibility.

**Requested Changes:**

* Inconsistent correlation values in Appendix D.3: The text states that MiLDEScore achieves a Spearman correlation of 0.908 with human evaluation, yet Table 6 reports a value of 0.88 under "Ours (Optimal)". Please correct this discrepancy and verify all related numbers.
* Optional: Add future work on inter-layer dependency reasoning: The current agent can lead to visual conflicts when edits to one layer (e.g., replacing a background) render another layer's content unreadable. I suggest adding a sentence in the Conclusion or Future Work section acknowledging the need for joint reasoning to achieve more harmonious design outcomes.

---

> ### Author Response · Authors · 2026-06-27
> **Official Response to Reviewer qctc**
>
> We sincerely thank the reviewer for the constructive comments and for recognizing the contributions of our work, including the proposed benchmark, evaluation protocol, and layer-aware editing framework. We address the reviewer’s concerns below.
>
> ## Weakness 1 / Requested Change 1: Inconsistent human correlation score.
>
> We thank the reviewer for carefully pointing out this inconsistency. The value reported in Appendix D.3 was a typo. The correct Spearman correlation between MiLDEScore and human evaluation is 0.88, consistent with Table 6, rather than 0.908. The value 0.908 came from an earlier intermediate version before all human evaluation results were collected, and we inadvertently forgot to update the corresponding text in Appendix D.3. We have corrected this number in the revised manuscript and have also verified all related correlation values to ensure consistency.
>
> ## Weakness 2 / Requested Change 2: Inter-layer editing conflicts.
>
> We agree with the reviewer that inter-layer dependency reasoning is an important limitation of our current framework. We also show the failure cases that come from this limitation in the appendix. As shown in Appendix H in the revised version (Appendix E in our original submission version), independently editing one layer can affect the visual compatibility of other layers after recomposition. To make this limitation clearer, we have added a new “Limitations and Future Work” section (Sec. 7 in the revised manuscript) at the end of the main paper, where we clarify the inter-layer editing conflicts as the second limitation of our work. We believe this is a valuable direction for future multi-layer document editing systems.

---

> ### Comment · Action_Editor_gXXp · 2026-07-12
> **Official recommendation**
>
> Dear Reviewer qctc,
>
> Could you please submit your official recommendation?
>
> Best,
>
> AC

---

> > ### Comment · Reviewer_qctc · 2026-07-20
> > **official comment**
> >
> > I have read the rebuttal and the revised manuscript. My concerns are resolved, I recommend this submission for acceptance.

---

### Author Response · Authors · 2026-06-27
**Official Response to All Reviewers and AC**

We thank all reviewers for their insightful and positive feedback. We are encouraged that the reviewers unanimously recognize the importance of our proposed dataset and methodologies. In our individual responses, we have addressed all concerns in detail. A revised manuscript incorporating these improvements has been uploaded, and we look forward to any further feedback.